# GraphMETRO: Mitigating Complex Graph Distribution Shifts via Mixture of Aligned Experts

**Shirley Wu**
Stanford University
shirwu@cs.stanford.edu

**Kaidi Cao**
Stanford University
kaidicao@cs.stanford.edu

**Bruno Ribeiro**[*]
Purdue University
ribeirob@purdue.edu

**James Zou**[*]
Stanford University
jamesz@cs.stanford.edu

**Jure Leskovec**[*]
Stanford University
jure@cs.stanford.edu

## Abstract

Graph data are inherently complex and heterogeneous, leading to a high natural diversity of distributional shifts. However, it remains unclear how to build machine learning architectures that generalize to the complex distributional shifts naturally occurring in the real world. Here, we develop GraphMETRO, a Graph Neural Network architecture that models natural diversity and captures complex distributional shifts. GraphMETRO employs a Mixture-of-Experts (MoE) architecture with a gating model and multiple expert models, where each expert model targets a specific distributional shift to produce a referential representation *w.r.t.* a reference model, and the gating model identifies shift components. Additionally, we design a novel objective that aligns the representations from different expert models to ensure reliable optimization. GraphMETRO achieves state-of-the-art results on four datasets from the GOOD benchmark, which is comprised of complex and natural real-world distribution shifts, improving by 67% and 4.2% on the WebKB and Twitch datasets. Code and data are available at https://github.com/Wuyxin/GraphMETRO.

## 1 Introduction

The intricate nature of real-world graph data introduces a wide variety of distribution shifts and heterogeneous graph variations [47, 30, 43, 26]. For instance, in a social graph, some user nodes can experience reduced activity and profile alterations, while other user nodes may see increased interactions. More broadly, such shifts go beyond the group-wise pattern and further contribute to the heterogeneous nature of graph data. In Figure 1, we provide a real-world example on a webpage network dataset, where, besides the general distribution shift from source to target distribution, two webpage nodes $u_1$ and $u_2$ in the target domain exhibit varying degrees of change in their content features. These inherent shifts and complexity accurately characterize the dynamics of real-world graph data, *e.g.,* social networks [2, 19] and ecommerce graphs [76].

Above the diverse graph variants, Graph Neural Networks (GNNs) [21, 25, 10] have become a prevailing method for downstream graph tasks. Standard evaluation often

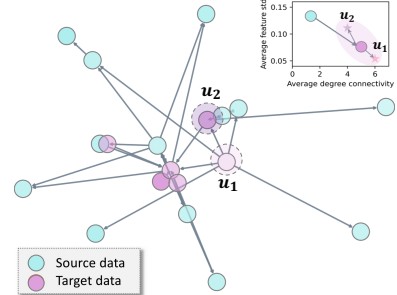

Figure 1: An example on WebKB [51, 20]. It illustrates (1) The distribution shift from source to target (the thick arrow in the upper right) and (2) Instance-wise heterogeneity in the target distribution (the thin arrows pointing to $u_1$ and $u_2$).

---

[*]Equal Senior Authorship

38th Conference on Neural Information Processing Systems (NeurIPS 2024).

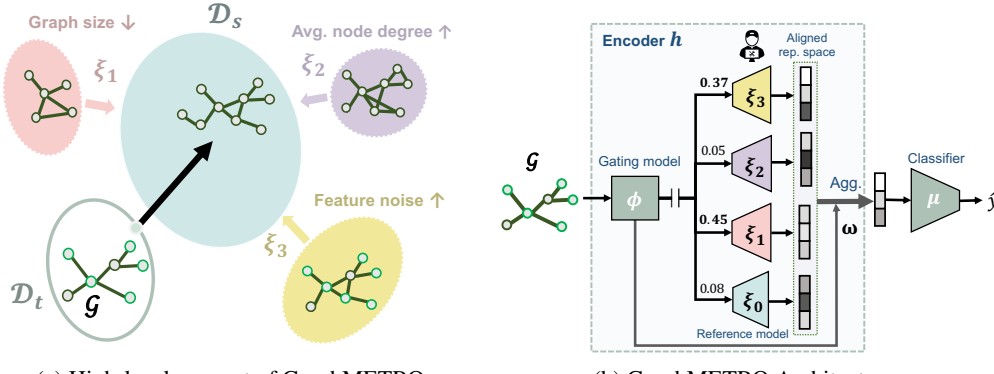

(a) High-level concept of GraphMETRO     (b) GraphMETRO Architecture

Figure 2: **Overview of GraphMETRO on graph classification tasks**. **(a) High-level Concept:** As a simple example, the distribution shift from a target graph $\mathcal{G} \in \mathcal{D}_t$ to a source distribution $\mathcal{D}_s$ is decomposed along three shift dimensions: graph size ($\xi_1$), node degree ($\xi_2$), and feature noise ($\xi_3$). Note that the shift components can be customized and expanded based on downstream tasks.
**(b) Architecture:** Given an input graph, the gating model $\mu$ decomposes the instance-specific distribution shift into the contributions from the shift components. Then, each expert model $\xi_i$ $(i > 0)$ is tasked with generating *referential invariant representations* (*cf.* Section 3 for the definition) *w.r.t.* an assigned shift component. $\xi_0$ is a reference model used for aligning the representation spaces of the expert models. The final representation is aggregated from the experts' output and is referentially invariant to any distribution shifts, which is then input to the classifier.

adopts random data splits for training and testing GNNs. However, it overlooks the complex distributional shifts naturally occurring in the real world. Moreover, compelling evidence shows that GNNs are extremely vulnerable to graph data shifts [79, 26, 20]. Thus, our goal is to build GNN models that generalize better to real-world data splits and graph dynamics described earlier.

Previous research on GNN generalization has mainly focused on two lines: (1) Data-augmentation training procedures that learn environment-robust predictors by augmenting the training data with the environment changes. For example, works have looked at distribution shifts related to graph size [49, 14], node features [26, 8, 27], and node degree or local structure [65, 39], assuming that the target data adhere to a designated shift type. (2) Learning environment-invariant representations or predictors either through inductive biases learned by the model [69, 67], through regularization [4, 32, 75], or a combination of both [72, 11, 80].

However, the real-world distribution shifts and graph dynamics are unknown. Specifically, the distribution shift could be any fusion of multiple shift dimensions, each characterized by unique statistical properties [26, 20, 50], which is rarely covered by single-dimension synthetic augmentation or fixed combinations of shift dimensions used in data augmentation approaches. Moreover, as seen in Figure 1, graph data may involve instance-wise heterogeneity, lacking stable properties from which invariant predictors can be learned [47, 30]. Here, the standard strategy of learning invariant predictors or representations must contend with a combinatorially large number of distribution shift variations. Thus, previous works may not be well-equipped to address this challenging task effectively.

Here we propose a novel and general framework, GraphMETRO. The key to our approach is to decompose any unknown shift into multiple shift components and learn predictors that can adapt to the graph heterogeneity observed in the target data. Figure 2a shows an example of our method on graph-level tasks, where the shift from the target graph data $\mathcal{G} \in \mathcal{D}_t$ to the source distribution $\mathcal{D}_s$ is decomposed into two strong shift components controlling feature noise and graph size, while the shift component controlling average node degree is identified as irrelevant. Specifically, the shift components are constructed such that each possesses unique statistical characteristics. Moreover, the contribution of each shift component to the shift is determined by an influence function that encodes the given graph $\mathcal{G}$ and source distribution $\mathcal{D}_s$. This design enables breaking down the generalization problem into (1) inference on strong shift components and their contributions as surrogates for any distributional or heterogeneous shifts, and (2) mitigation toward the surrogate shifts, where the individual shift components are interpretable and more tractable.

For the first subproblem, we design a hierarchical architecture composed of a gating model and multiple expert models, inspired by the mixture-of-experts (MoE) architecture [24]. As shown in Figure 2b, the gating model takes any given node or graph data and identifies strong shift components that govern the localized distribution shift, while each expert model corresponds to an individual shift component. Second, to further mitigate surrogate distribution shifts, we train the expert models to generate referentially invariant representations with respect to their corresponding shift components, which are then aggregated as the final representation vector. Moreover, the expert outputs must align properly in a common representation space to prevent extreme divergence in the aggregated representation. Consequently, we design a novel objective to ensure a smooth training process. Finally, during the evaluation process, we integrate outputs from both the gating and expert models for final representations.

This process effectively generates invariant representations across complex distributional shifts. To highlight, our method achieves the best performance on four node- and graph-level tasks from the GOOD benchmark [20], which involves a diverse set of natural distribution shifts such as user language shifts in gamer networks and university domain shifts in university webpage networks. GraphMETRO achieves a 67% relative improvement over the state-of-the-art on the WebKB dataset [51]. On synthetic datasets, our method outperforms Empirical Risk Minimization (ERM) by 4.6% on average. To the best of our knowledge, GraphMETRO is the first to explicitly target complex distribution shifts that resemble real-world settings.

The key benefits of GraphMETRO are as follows:

- **A novel paradigm:** GraphMETRO provides a new approach to aid GNN generalization by decomposing and mitigating complex distributional shifts via a mixture-of-experts architecture.
- **Superior performance:** It outperforms state-of-the-art methods on real-world datasets with natural splits and shifts, demonstrating promising generalization ability.
- **Enhanced interpretability:** GraphMETRO offers insights into the shift types of graph data by identifying and interpreting strong shift components.

## 2   Related Works

**Invariant learning for graph OOD**. The prevailing invariant learning approaches assume that there exist an underlying graph structure (*i.e.,* subgraph) [69, 36, 34, 71, 60, 83, 37] or representation [1, 67, 6, 3, 81, 68, 11, 8, 42] that is invariant to different environments and/or causally related to the label of a given instance. For example, DIR [69] constructs interventional distributions and distills causal subgraph patterns to make generalizable predictions for graph-level tasks. However, this line of research focuses on group patterns without explicitly considering instance heterogeneity. Therefore, the standard invariant learning approaches are not well-equipped to mitigate the complex distribution shifts in our context. See Appendix A for an in-depth comparison.

**Data augmentation for graph OOD**. GNNs demonstrate robustness to data perturbations when incorporating augmented views of graph data [7]. Previous works have explored augmentation with respect to graph sizes [85, 4, 84], local structures [40, 38], feature metrics [15], and graphons[33]. For example, OOD-G-Mixup [22] creates virtual OOD samples by perturbing the graph rationale space. Recently, Jin et al. [23] proposed adapting testing graphs to transformed graphs with patterns similar to the training graphs. Other approaches conduct augmentation implicitly via attention mechanisms [45, 66]. For example, GSAT [45] injects stochasticity into attention weights to block label-irrelevant information. Nevertheless, this line of research may not effectively solve the challenging problem, since unseen distribution shifts may not be covered by the distribution of augmented graphs. Moreover, it may lead to degradation of in-distribution performance due to GNNs' limited expressiveness in encoding a broad distribution.

**Instance heterogeneity for graph OOD**. Recent methods [41, 59, 61, 77, 35, 74] have explicitly considered instance heterogeneity for improving OOD generalization in GNNs. For example, OOD-GNN [35] mitigates instance-wise heterogeneity by eliminating spurious correlations between irrelevant and relevant graph representations through nonlinear decorrelation and sample reweighting. Yao et al. [74] focus on explicitly model domain correlations and spurious features and adapt to each test instance's unique distribution shifts. While these methods explicitly consider instance heterogeneity in graph OOD problems, they often focus on specific types of distribution shifts or rely on the assumption that target data adhere to certain designated shift types.

In contrast, our method introduces a novel paradigm that decomposes any unknown shift into multiple shift components and learns predictors that can adapt to the graph heterogeneity observed in the target data. By leveraging a mixture-of-experts architecture, our approach can handle complex distribution shifts without assuming specific shift types or relying solely on group patterns.

**Mixture-of-expert models**. The applications of mixture-of-expert models (MoE) [24, 57] have largely focused on their efficiency and scalability [13, 12, 53, 9], particularly in image and language domains. For image domain generalization, Li et al. [31] focus on neural architecture design and integrate expert models with vision transformers to capture correlations in the training dataset that may benefit generalization, where an expert is responsible for a group of similar visual attributes. Puigcerver et al. [52] observed improved robustness by adopting MoE models in the image domain. In the graph domain, differently motivated from our work, Wang et al. [64] consider experts as information aggregation models with varying hop sizes to capture different ranges of message passing, aiming to improve model expressiveness on large-scale data.

GraphMETRO is the first to design a mixture-of-expert model specifically tailored to address complex distribution shifts in graphs, coupled with a novel objective for producing invariant representations. While previous methods mostly focus on either node- or graph-level tasks, GraphMETRO is a more general solution applicable to both.

## 3 Method

**Problem formulation**. For simplicity, we consider a graph classification task and later extend it to node-level tasks. Let $\mathcal{D}_s$ be the source distribution and $\mathcal{D}_t$ be an unknown target distribution. We are interested in the natural graph distribution shifts. Our goal is to learn a model $f_\theta$ with high generalization ability. The standard approach is Empirical Risk Minimization (ERM), *i.e.,*

$$\theta^* = \arg \min_{\theta} \ \mathbb{E}_{(\mathcal{G},y) \sim \mathcal{D}_s} \mathcal{L}\left(f_\theta\left(\mathcal{G}\right), \ y\right), \tag{1}$$

where $\mathcal{L}$ denotes the loss function and $y$ is the label of the graph $\mathcal{G}$. However, the assumptions underlying ERM can be easily violated, making $\theta^*$ suboptimal. Moreover, since the distribution shift is unknown and cannot provide supervision for model training, the direct optimization of Eq 1 is intractable.

### 3.1 Shift Components

Based on the common mixture pattern studied in real-world networks [29, 30, 50], we propose the following informal assumption:

**Assumption 1 (An equivalent mixture for distribution shifts)** *Let the distribution shift between the source $\mathcal{D}_s$ and target $\mathcal{D}_t$ distributions be the result of an unknown intervention in the graph formation mechanism. We assume that the resulting shift in $\mathcal{D}_t$ can be modeled by up to $k$ out of $K$ classes of stochastic transformations applied to each instance in the source distribution $\mathcal{D}_s$ ($k \leq K$).*

Assumption 1 essentially states that any distribution shift can be decomposed into $k$ shift components of stochastic graph transformations. The assumption simplifies the generalization problem by enabling the modeling of individual shift components that constitute the shift and their respective contributions to the overall distribution shift. While this assumption is generally applicable, as observed in the experiments, we include a discussion on scenarios that fall outside the scope of this assumption in Appendix F. Previous works [28, 69, 67] implicitly infer such shift components from the data environments constructed based on the source distribution. However, distilling diverse shift components from the source data is challenging due to the complexity of the graph distribution shifts and largely depends on the constructed environments[2].

**Graph extrapolation as shift components**. To construct the shift components, we employ a data extrapolation technique based on the source data. In particular, we introduce $K$ independent classes of transform functions, including multihop subgraph sampling, the addition of Gaussian feature

---

[2]In other words, if the distribution shifts were described via environment assignments, one would have a combinatorial number of such environments, *i.e.,* the product of all different subsets of nodes and all their possible distinct shifts.

noise, and random edge removal [54]. The $i$-th class, governed by the $i$-th shift component, defines a stochastic transformation $\tau_i$ that transforms an input source graph $\mathcal{G}$ into an output graph $\tau_i(\mathcal{G})$, where $i = 1, \ldots, K$. For instance, $\tau_i$ can be defined to randomly remove edges with an edge-dropping probability in the range of $[0.3, 0.5]$. Note that the extrapolation aims to construct the basis of shifts rather than directly conducting data augmentation, as explained in Eq 3 later.

## 3.2 Mixture of Aligned Experts

In light of the shift components, we formulate the generalization problem as two separate phases:

- **Surrogate estimation:** Identify a mixture of shift components as the surrogate for the target shifts, where the mixture can vary across different node or graph instances to capture heterogeneity.

- **Mitigation and aggregation:** Mitigate individual shift components, followed by aggregating the representations output by each expert to resolve the surrogate shift.

**Overview**. Inspired by the mixture-of-experts (MoE) architecture [24], the core idea of GraphMETRO is to build a hierarchical architecture composed of a gating model and multiple expert models, where the gating model predicts the influence of the shift components on a given instance. For the expert models, we design each to handle an individual shift component. The experts produce referential representations invariant to their designated shift component, with the representations aligned in a common representation space. Finally, our architecture combines the expert outputs into a final representation, which our training objective ensures is invariant to the stochastic transformations within the mixture distribution. We detail each module as follows:

**Gating model**. We introduce a GNN $\phi$ as the gating model, which takes any graph as input and outputs a weight vector $\boldsymbol{w}$ on the shift components. The weight vector suggests the most probable shift components from which the input graph originates. For example, in Figure 2b, given an unseen graph with decreased graph size and node feature noise, a trained gating model should assign large weights to the corresponding shift components and small values to the irrelevant ones. Note that $\phi$ should be such that $\boldsymbol{w}_i$, the weight on the $i$-th component, strives to be sensitive to the stochastic transformation $\tau_i$ but insensitive to the application of other stochastic transformations $\tau_j, j \neq i$. This way, determining whether the $i$-th component is present should not depend on other components.

**Expert models**. We build $K$ expert models, each corresponding to a shift component. Formally, we denote an expert model as $\xi_i : \mathcal{G} \to \mathbb{R}^v$, where $v$ is the hidden dimension, and we use $\mathbf{z}_i = \xi_i(\mathcal{G})$ to denote the output representation. Each expert model essentially produces invariant representations [48] with respect to the distribution shift controlled by its assigned shift component. However, independently optimizing each expert without properly aligning the expert's output space is incompatible with model training. Specifically, an expert model may learn its own unique representation space, which may cause information loss when its output is aggregated with other expert outputs. Moreover, aggregating independent representations results in a mixed representation space with high variance, which makes it difficult for the predictor head, such as multi-layer perceptrons (MLPs), to capture the interactions and dependencies among these diverse representations and output rational predictions. Thus, aligning the representation spaces of experts is necessary to ensure compatibility and facilitate stable model training. To align the experts' output spaces properly, we introduce the concept of referential invariant representation:

**Definition 1 (Referential Invariant Representation)** *Let $\mathcal{G}$ be an input graph and let $\tau$ be an arbitrary stochastic transform function, with domain and co-domain in the space of graphs. Let $\xi_0$ be a model that encodes a graph into a representation. A referential invariant representation w.r.t. the given $\tau$ is denoted as $\xi^*(\mathcal{G})$, where $\xi^*$ is a function that maps the original data $\mathcal{G}$ to a high-dimensional representation $\xi^*(\mathcal{G})$ such that $\xi_0(\mathcal{G}) \approx \xi^*(\tau(\mathcal{G}))$ holds for every $\mathcal{G} \in \mathrm{supp}(\mathcal{D}_s)$, where $\mathrm{supp}(\mathcal{D}_s)$ denotes the support of $\mathcal{D}_s$. We refer to $\xi_0$ as the reference model.*

Thus, the representation space of the reference model serves as an intermediate to align different experts, while each expert $\xi_i$ has its own ability to produce referential invariant representations *w.r.t.* a stochastic transform function $\tau_i, i = 1, \ldots, K$. We include the reference model as a special "in-distribution" expert model on the source data.

**Architecture design for the expert models**. Further, we propose two architecture designs for the expert models. A straightforward way is to construct $(K + 1)$ GNN encoders to generate referential

invariant representations for individual shift components. This ensures model expressiveness while increasing memory usage due to multiple encoders. To alleviate this concern, we provide an alternative approach. Specifically, we can construct a shared module, *e.g.,* a GNN encoder, among the expert models, coupled with a specialized module, *e.g.,* an MLP, for each expert. We discuss the impact of architecture choices on model performance in the experiment section.

**The MoE workflow**. Given a node or graph instance, the gating model assigns weights $\boldsymbol{w} \in \mathbb{R}^{K+1}$ over the expert models, indicating the mixture of shift components on the instance. The output weights, being conditional on the input instance, enable the depiction of heterogeneous distribution shifts that vary across instances. After that, we obtain the output representations from the expert models, which eliminate the effect of the corresponding shift component. Then, the final representation is computed via aggregating the representations based on the weight vector, *i.e.,*

$$h(\mathcal{G}) = \text{Aggregate}(\{(\phi(\mathcal{G})_i, \xi_i(\mathcal{G})) \mid i = 0, 1, \ldots, K\})$$

where $h$ is the encoder of $f$. The aggregation function can be a weighted sum over the expert outputs or a selection function that selects the expert output with maximum weight, *e.g.,*

$$h(\mathcal{G}) = \text{Softmax}(\boldsymbol{w}) \cdot [\mathbf{z}_0, \ldots, \mathbf{z}_K]^T \tag{2}$$

Assuming the distribution shift on an instance is controlled by any single shift component, we have $h(\tau_i(\mathcal{G})) = \xi_i(\tau_i(\mathcal{G})) \approx \xi_0(\mathcal{G}) = h(\mathcal{G})$ for $i = 0, \ldots, K$, where $\xi_i(\tau_i(\mathcal{G})) \approx \xi_0(\mathcal{G})$ holds according to Definition 1. This indicates that $h$ automatically produces referentially invariant representations while allowing heterogeneity across different instances, *e.g.,* different shift types or control strengths. For clarity, we define $\tau^{(k)}$ as a joint stochastic transform function composed of any $k$ or fewer transform functions out of the $K$ transform functions. We refer to the scenario where $h$ produces referentially invariant representations *w.r.t.* $\tau^{(k)}$ as $\tau^{(k)}$-invariance. To extend $k$ to higher orders ($k > 1$), we design the objective in Section 3.3, which enforces $h$ to satisfy $\tau^{(k)}$-invariance, ensuring model generalization when multiple shifts exist. After that, a classifier $\mu$ takes the aggregated representation from Eq 2 for prediction tasks. Thus, we have $f = \mu \circ h$ as the mixture-of-experts model.

### 3.3 Training Objective

As shown in Figure 2b, we consider three trainable modules, *i.e.,* the gating model $\phi$, the expert models $\{\xi_i\}_{i=0}^K$, and the classifier $\mu$. We propose the following objective:

$$\min_\theta \mathcal{L}_f = \min_\theta(\mathcal{L}_1 + \mathcal{L}_2), \text{ where}$$
$$\mathcal{L}_1 = \mathbb{E}_{(\mathcal{G}, y) \sim \mathcal{D}_s} \mathbb{E}_{\tau^{(k)}} \text{BCE}(\phi(\tau^{(k)}(\mathcal{G})), Y(\tau^{(k)})) \tag{3}$$
$$\mathcal{L}_2 = \mathbb{E}_{(\mathcal{G}, y) \sim \mathcal{D}_s} \mathbb{E}_{\tau^{(k)}} [\text{CE}(\mu(h(\tau^{(k)}(\mathcal{G})), y)) + \lambda \cdot d(h(\tau^{(k)}(\mathcal{G})), \xi_0(\mathcal{G}))]$$

- $\mathcal{L}_1$: $Y(\tau^{(k)}) \in \{0, 1\}^{K+1}$ is the ground truth vector, and its $i$-th element is 1 if and only if $\tau_i$ composes $\tau^{(k)}$. BCE is the Binary Cross Entropy. This term indicates that the gating model $\phi$ is optimized to accurately predict a mixture of shift components.

- $\mathcal{L}_2$: CE is the Cross Entropy function. $d(\cdot, \cdot)$ is a distance function between two representations, and $\lambda$ is a parameter controlling the strength of the distance penalty. In the experiments, we use the Frobenius norm as the distance function, *i.e.,* $d(\mathbf{z}_1, \mathbf{z}_2) = \frac{1}{n}\|\mathbf{z}_1 - \mathbf{z}_2\|_F = \frac{1}{n}\sqrt{\sum_{i=1}^n (\mathbf{z}_{1i} - \mathbf{z}_{2i})^2}$, and we use $\lambda = 1$ for all the experiments. The second loss term optimizes the expert models and the classifier, and we prevent it from backpropagating to the gating model to avoid interference. Specifically, $\mathcal{L}_2$ aims to improve the encoder's performance in predicting graph classes and achieves referential alignment with the reference model $\xi_0$ via the distance function. Note that, when $k > 1$, $\mathcal{L}_2$ also enforces $h$ to be invariant to multiple shifts via the $\tau^{(k)}$-invariance condition.

We optimize our model via stochastic gradient descent, where $\tau^{(k)}$ is sampled at each gradient step. Overall, GraphMETRO yields a MoE model, comprising a gating model with high predictive accuracy, expert models that are aligned and can generate invariant representations in a shared representation space, and a task-specific classifier that utilizes robust and invariant representations for class prediction.

Table 1: **Test results on the real-world datasets.** We compute the p-value between the results of GraphMETRO and the state-of-the-art methods. The results of GraphMETRO is repeated five times.

| | Node classification | | Graph classification | | Require domain information |
|---|---|---|---|---|---|
| | WebKB | Twitch | Twitter | SST2 | |
| ERM | 14.29 ± 3.24 | 48.95 ± 3.19 | 56.44 ± 0.45 | 80.52 ± 1.13 | No |
| DANN | 15.08 ± 0.37 | 48.98 ± 3.22 | 55.38 ± 2.29 | 80.53 ± 1.40 | No |
| IRM | 13.49 ± 0.75 | 47.21 ± 0.98 | 55.09 ± 2.17 | 80.75 ± 1.17 | Yes |
| VREx | 14.29 ± 3.24 | 48.99 ± 3.20 | 55.98 ± 1.92 | 80.20 ± 1.39 | Yes |
| GroupDRO | 17.20 ± 0.76 | 47.20 ± 0.44 | 56.65 ± 1.72 | 81.67 ± 0.45 | Yes |
| Deep Coral | 13.76 ± 1.30 | 49.64 ± 2.44 | 55.16 ± 0.23 | 78.94 ± 1.22 | Yes |
| SRGNN | 13.23 ± 2.93 | 47.30 ± 1.43 | NA | NA | Yes |
| EERM | 24.61 ± 4.86 | 51.34 ± 1.41 | NA | NA | No |
| OODGAT | 14.41 ± 1.10 | 49.38 ± 0.87 | NA | NA | |
| DIR | NA | NA | 55.68 ± 2.21 | 81.55 ± 1.06 | No |
| G-Mixup | NA | NA | 53.32 ± 2.75 | 77.43 ± 1.97 | |
| GSAT | NA | NA | 56.40 ± 1.76 | 81.49 ± 0.76 | No |
| CIGA | NA | NA | 55.70 ± 1.39 | 80.44 ± 1.24 | No |
| GraphMETRO | **41.11 ± 7.47** | **53.50 ± 2.42** | **57.24 ± 2.56** | **81.87 ± 0.22** | No |
| p-value | **< 0.001** | **0.023** | **0.042** | **0.081** | - |

## 3.4 Discussion and Analysis

**Node classification tasks**. While we introduce our method following a graph-level task setting, GraphMETRO is readily adaptable for node-level tasks. Instead of generating graph representations, GraphMETRO is capable of producing node-level invariant representations. Additionally, we apply stochastic transform functions to the subgraph containing a target node and identify its shift components, which is consistent with the objective in Equation 3.

**Interpretability**. The gating model of GraphMETRO predicts the shift components on the node or graph instance, which provides interpretations and insights into the distribution shifts in unknown datasets. In contrast, existing research on GNN generalization [69, 45, 6, 67] often lacks proper identification and analysis of distribution shifts prevalent in real-world datasets. This creates a gap between human understanding of graph distribution shifts and the actual graph dynamics. To bridge this gap, we offer an in-depth study of the experiments to demonstrate GraphMETRO ' insights into the complexity of real graph distributions.

**Computational cost**. The forward process of $f$ requires $O(K)$ encoder passes, using the weighted sum aggregation from $(K + 1)$ expert outputs. Since the extrapolation process increases the dataset size by a factor of $(K + 1)$, the training computation complexity is $O(K^2|\mathcal{D}_s|)$, where $|\mathcal{D}_s|$ is the size of the source dataset.

## 4 Experiments

We perform systematic experiments on both real-world (Section 4.1) and synthetic datasets (Section 4.2) to validate the generalizability of GraphMETRO under complex distribution shifts.

### 4.1 Applying GraphMETRO to Real-world Datasets

We perform experiments on real-world datasets, which introduce complex and natural distribution shifts. In these scenarios, the test distribution may not precisely align with the mixture mechanism encountered during training.

**Datasets**. We use four classification datasets, *i.e.,* WebKB [51], Twitch [55], Twitter [78], and GraphSST2 [78, 58], using the dataset splits from the GOOD benchmark [20], which exhibit various real-world covariate shifts. Specifically, WebKB is a 5-class prediction task that predicts the classes of university webpages, with nodes split based on different university domains, demonstrating a natural challenge of applying GNNs trained on some university data to other unseen data. Twitch is a binary classification task that predicts whether a user streams mature content, with nodes split mainly by user language domains. Twitter and GraphSST2 are real-world grammar tree graph datasets,

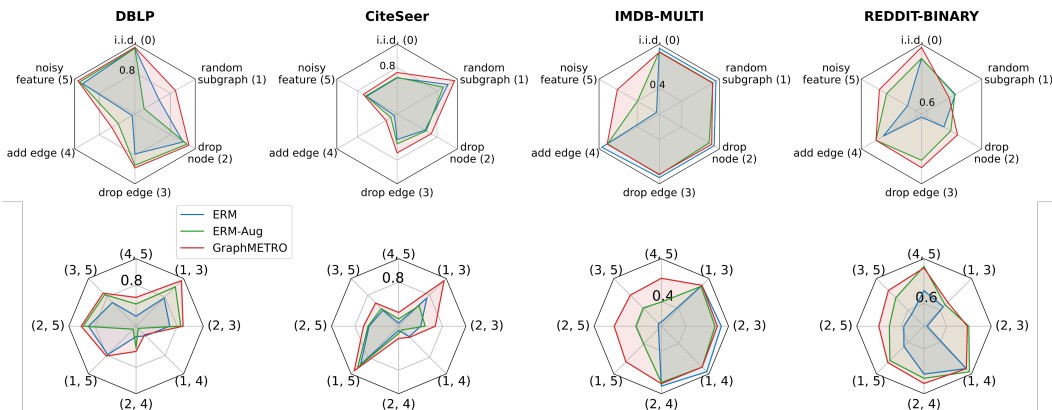

Figure 3: **Accuracy on synthetic distribution shifts**. The first row shows the testing accuracy on single shift components. We label the distribution by the clockwise order. The second row shows the testing accuracy on distribution shifts with multiple shift components, where each testing distribution is a composition of two different transformations. For example, *(1, 5)* denotes a testing distribution where each graph is controlled by *random subgraph (1)* and *noisy feature (5)* shift components. We include the numerical values in Appendix E.

where graphs from different domains differ in sentence length and language style, posing a direct challenge of generalizing to different language lengths, styles, and contexts.[3]

**Baselines**. We use ERM and domain generalization baselines, including DANN [17], IRM [1], VREx [28], GroupDRO [56], and Deep Coral [62]. Moreover, we compare GraphMETRO with robustness/generalization techniques for GNNs, including DIR [69], OODGAT [59], GSAT [45], and CIGA [6] for graph classification tasks, and SR-GCN [85], EERM [67], and G-Mixup [22] for node classification tasks.

**Training and evaluation**. We use an individual GNN encoder for each expert in the experiments. Additionally, we include the results of using a shared module among experts in Appendix D.1 due to space limitations. For evaluation metrics, we use ROC-AUC on Twitch and classification accuracy on the other datasets following [20]. See Appendix B for details about the architectures and optimizer.

**Results**. In Table 1, we observe that GraphMETRO consistently outperforms the baseline models across all datasets. It achieves notable improvements of 67.0% and 4.2% relative to EERM on the WebKB and Twitch datasets, respectively. When applied to graph classification tasks, Graph-METRO shows significant improvements, as the baseline methods exhibit similar performance levels. Importantly, GraphMETRO can be applied to both node- and graph-level tasks, whereas many graph-specific methods designed for generalization are limited to one of these tasks. Additionally, GraphMETRO does not require any domain-specific information during training.

**Main Conclusion**. The observation that GraphMETRO is the best-performing method demonstrates its significance for real-world applications, as it excels in handling unseen and wide-ranging distribution shifts. This adaptability is crucial, as real-world graph data often exhibit unpredictable shifts that can affect model performance. Thus, GraphMETRO ' versatility ensures its reliability across diverse domains, safeguarding performance in complex real-world scenarios. In Appendix D.2 and D.3, we provide two studies on the impact of the alignment term controlled by $\lambda$ and the stochastic transform function choices on the model performance, analyzing the sensitivity and success of GraphMETRO .

## 4.2 Inspect GraphMETRO on Synthetic Datasets

Following the experiments on real-world datasets, we perform experiments on synthetic datasets to further inspect and validate the effectiveness of our approach.

---

[3]We specifically exclude datasets with synthetic shifts from the GOOD benchmark. We leave the applications to molecular datasets in the GOOD benchmark for future work, as it requires designing shift components based on expert knowledge.

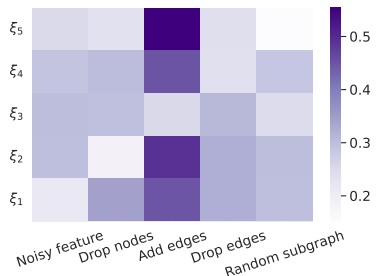

(a) **Invariance matrix on Twitter dataset**

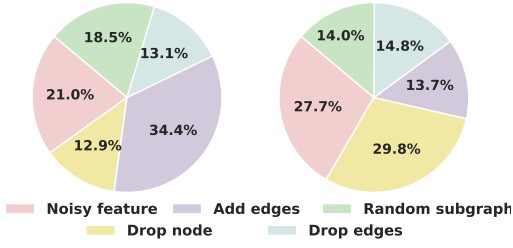

(b) **Mixture of distribution shifts** on WebKB (left) and Twitch (right) identified by GraphMETRO.

Figure 4: (a) Invariance matrix on the Twitter dataset. Lighter colors indicate a higher invariance of representations produced by each expert. Small values on the diagonal elements of the invariance matrix indicate that each expert excels at generating invariant representations *w.r.t.* the specific shift component. (b) Mixture of distribution shifts identified by GraphMETRO. Higher values indicate a strong shift component in the testing distribution.

**Datasets**. We use graph datasets from citation and social networks. For node classification tasks, we use DBLP [16] and CiteSeer [73]. For graph classification tasks, we use REDDIT-BINARY and IMDB-MULTI [46]. See Appendix B for dataset processing and details of the transform functions.

**Training and evaluation**. We adopt the same encoder architecture for Empirical Risk Minimization (ERM), ERM with data augmentation (ERM-Aug), and the expert models of GraphMETRO. For ERM-Aug training, we augment the training datasets using the same transform functions we used to construct the testing environments. Finally, we select the model based on the in-distribution validation accuracy and report the testing accuracy on each environment from five trials. See Appendix B for detailed settings and hyperparameters.

**Results**. Figure 3 illustrates our model's performance across single (the first row) and multiple (the second row) shift components. In most test distributions, GraphMETRO exhibits significant improvements or performs on par with two other methods. Notably, on the IMDB-MULTI dataset with noisy node features, GraphMETRO outperforms ERM-Aug by 5.9%, and it enhances performance on DBLP by 4.4% when dealing with random subgraph sampling. In some instances, GraphMETRO even demonstrates improved results on in-distribution datasets, such as a 2.9% and 2.0% boost on Reddit-BINARY and DBLP, respectively. This could be attributed to the increased model expressiveness of the MoE architecture or weak distribution shifts that can exist in the randomly split testing datasets.

### 4.3 Invariance Matrix for Inspecting GraphMETRO

A key insight from GraphMETRO is that each expert excels in generating invariant representations specifically for a shift component. To delve into the modeling mechanism, we denote $I \in \mathbb{R}^{K \times K}$ as an invariance matrix. This matrix quantifies the sensitivity of expert $\xi_i$ to the $j$-th shift component. Specifically, for $i \in [K]$ and $j \in [K]$, we have

$$I_{ij} = \mathbb{E}_{\mathcal{G} \sim \mathcal{D}_s} \mathbb{E}_{\tau_j} [d(\xi_i(\tau_j(\mathcal{G})), \xi_0(\mathcal{G}))]$$

Ideally, for a given shift component, the representation produced by the corresponding expert should be most similar to the representation produced by the reference model. That is, the diagonal entries $I_{ii}$ should be smaller than the off-diagonal entries $I_{ij}$ for $j \neq i$ and $i = 1, \ldots, K$. In Figure 4a, we visualize the normalized invariance matrix computed for the Twitter dataset, revealing a pattern that aligns with the analysis. This demonstrates that GraphMETRO effectively adapts to various distribution shifts, indicating that our approach generates consistent invariant representations for each of the shift components.

### 4.4 Distribution Shift Discovery

With the trained MoE model, we aim to understand the distribution shifts in the target distribution. Here we conduct case studies on the WebKB and Twitch datasets. Specifically, we first validate the

gating models' ability to identify mixtures, which is a multitask binary classification with $(K + 1)$ classes. The gating models achieve high accuracies of 92.4% on WebKB and 93.8% on the Twitch dataset. As mixtures output by gating models identify significant shift components on an instance, we leverage them as human-understandable interpretations and compute the average mixture across $\mathcal{G} \in \mathcal{D}_t$ as the global mixture on the target distribution. The results in Figure 4b show that the shift component, increased edges, dominates on the WebKB dataset, while the shift components controlling, *e.g.,* node features and decreasing nodes, show large effects on the Twitch dataset. The results align with dataset structures, *i.e.,* WebKB's natural shifts across different university domains and Twitch's language-based shifts. While quantitatively validating these observations in complex graph distributions remains a challenge, we aim to explore these complexities in greater depth in future work, which can potentially offer insights into real-world graph dynamics.

## 5    Conclusion and Future Work

This work mitigates the challenge of improving the generalization of Graph Neural Networks (GNNs) to real-world data splits and dynamic graph distributions. To tackle these shifts, we introduce **GraphMETRO**, a mixture-of-aligned-experts architecture, which models graph distribution shifts as mixtures of shift components, each controlling shifts in unique directions with varying complexity.

GraphMETRO distinguishes itself from traditional invariant learning methods, which often rely on environment variables to partition data. Instead, our method treats distribution shifts as mixtures, represented by the gating function's score vector, allowing for infinite environments due to the continuous nature of the score. When restricted to binary outputs, GraphMETRO can simulate finite environments, making it flexible and versatile. Furthermore, the introduction of referential invariant representation via a reference model is a key innovation of our approach.

Experimental results demonstrate that GraphMETRO consistently outperforms baseline methods on real-world datasets, achieving significant improvements. Additional synthetic studies and case analyses further validate the method's effectiveness and adaptability across diverse scenarios.

In future work, we aim to explore the broader applicability of GraphMETRO , including potential extensions to address label distributional shifts. Detailed discussions on these directions are provided in Appendix F.

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

# A Theoretical Analysis

We provide a theoretical justification for why GraphMETRO can effectively address complex graph distribution shifts and outperform existing approaches. Our analysis focuses on three key aspects: (1) the limitations of existing methods, (2) how GraphMETRO overcomes these limitations, and (3) the theoretical guarantees of our approach.

## A.1 Limitations of Existing Approaches

Consider a graph classification task with the following causal model:

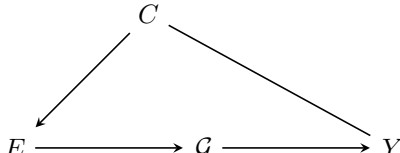

where $\mathcal{G}$ is the input graph, $Y$ is the label, $E$ is an unobserved environment variable, and $C$ is an unobserved confounder.

Existing approaches primarily focus on learning environment-invariant predictors $f(\mathcal{G})$ such that:

$$P(Y|f(\mathcal{G}), E = e) \approx P(Y|f(\mathcal{G})), \quad \forall e \in \mathbb{E} \tag{4}$$

However, these methods face significant challenges when:

- The environment space $\mathbb{E}$ is vast and complex.
- The distribution shifts are heterogeneous across instances.
- The shifts involve multiple interacting components.

## A.2 GraphMETRO's Approach

GraphMETRO addresses these limitations through its novel architecture and training objective:

1. **Decomposition of shifts**: Instead of learning a single invariant predictor, GraphMETRO decomposes the complex shift into $K$ shift components:

$$E = (E_1, E_2, ..., E_K) \tag{5}$$

where each $E_i$ represents a specific type of graph transformation.

2. **Mixture-of-Experts**: The gating model $\phi$ and expert models $\{\xi_i\}_{i=0}^{K}$ allow for adaptive handling of heterogeneous shifts:

$$h(\mathcal{G}) = \sum_{i=0}^{K} w_i(\mathcal{G}) \cdot \xi_i(\mathcal{G}) \tag{6}$$

where $w_i(\mathcal{G}) = \phi(\mathcal{G})_i$ are instance-dependent weights.

3. **Referential alignment**: The training objective enforces alignment between expert outputs and a reference model:

$$\mathcal{L}_2 = \mathbb{E}_{(\mathcal{G},y)\sim\mathcal{D}_s}\mathbb{E}_{\tau^{(k)}}[\text{CE}(\mu(h(\tau^{(k)}(\mathcal{G})), y)) + \lambda \cdot d(h(\tau^{(k)}(\mathcal{G})), \xi_0(\mathcal{G}))] \tag{7}$$

## A.3 Theoretical Guarantees

We now provide theoretical guarantees for GraphMETRO's performance:

**Theorem 1 (Shift-Invariance)** *For any graph $\mathcal{G}$ and shift component $\tau_i$, the encoder $h$ satisfies:*

$$h(\tau_i(\mathcal{G})) = h(\mathcal{G}) \tag{8}$$

**Proof 1** *Given the gating model's ability to identify shift components and the expert models' invariance properties:*

$$h(\tau_i(\mathcal{G})) = \sum_{j=0}^{K} w_j(\tau_i(\mathcal{G})) \cdot \xi_j(\tau_i(\mathcal{G})) \tag{9}$$

$$= w_i(\tau_i(\mathcal{G})) \cdot \xi_i(\tau_i(\mathcal{G})) \tag{10}$$

$$= w_i(\mathcal{G}) \cdot \xi_0(\mathcal{G}) \tag{11}$$

$$= h(\mathcal{G}) \tag{12}$$

*where the second equality holds because the gating model identifies $\tau_i$, and the third equality follows from the definition of referential invariant representation.*

This theorem guarantees that GraphMETRO can handle individual shift components. We can extend this to combinations of shifts:

**Theorem 2 (Composition of Shifts)** *For any graph $\mathcal{G}$ and combination of $k$ shift components $\tau^{(k)} = \tau_{i_1} \circ \tau_{i_2} \circ ... \circ \tau_{i_k}$, the encoder $h$ approximately satisfies:*

$$h(\tau^{(k)}(\mathcal{G})) \approx h(\mathcal{G}) \tag{13}$$

**Proof 2** *We prove this theorem by induction on $k$, the number of shift components.*

*Base case ($k = 1$): This is directly given by Theorem 1.*

*Inductive step: Assume the theorem holds for $k-1$ shift components. We need to prove it holds for $k$ shift components.*

*Let $\tau^{(k)} = \tau_{i_k} \circ \tau^{(k-1)}$ where $\tau^{(k-1)} = \tau_{i_1} \circ \tau_{i_2} \circ ... \circ \tau_{i_{k-1}}$.*

$$h(\tau^{(k)}(\mathcal{G})) = h(\tau_{i_k}(\tau^{(k-1)}(\mathcal{G}))) \tag{14}$$

$$\approx h(\tau^{(k-1)}(\mathcal{G})) \quad \text{(by Theorem 1)} \tag{15}$$

$$\approx h(\mathcal{G}) \quad \text{(by induction hypothesis)} \tag{16}$$

*The approximation in the second line comes from the fact that the gating model $\phi$ may not perfectly identify the shift component $\tau_{i_k}$ when applied after $\tau^{(k-1)}$. However, our training objective $\mathcal{L}_2$ explicitly minimizes:*

$$\mathbb{E}_{\tau^{(k)}}[d(h(\tau^{(k)}(\mathcal{G})), \xi_0(\mathcal{G}))] \tag{17}$$

*This ensures that even for compositions of shifts, the output of $h$ remains close to the reference model $\xi_0$, which is invariant to all shifts.*

*Therefore, by induction, the theorem holds for any $k \geq 1$.*

**Theorem 3 (Generalization Bound)** *Let $\mathcal{L}(\cdot, \cdot)$ be the cross-entropy loss. For any distribution $\mathcal{D}_t$ resulting from a combination of shift components in $\tau^{(k)}$, the generalization error of GraphMETRO satisfies:*

$$\mathbb{E}_{(\mathcal{G},y)\sim\mathcal{D}_t}[\mathcal{L}(f(\mathcal{G}), y)] \leq \mathbb{E}_{(\mathcal{G},y)\sim\mathcal{D}_s}[\mathbb{E}_{\tau^{(k)}}[\mathcal{L}(f(\tau^{(k)}(\mathcal{G})), y)]] + \epsilon \tag{18}$$

*where $\epsilon$ is a small constant depending on the complexity of the model and the number of samples.*

**Proof 3** *1) Our training objective minimizes:*

$$\mathcal{L}_{train} = \mathbb{E}_{(\mathcal{G},y)\sim\mathcal{D}_s}[\mathbb{E}_{\tau^{(k)}}[\mathcal{L}(f(\tau^{(k)}(\mathcal{G})), y)]] \tag{19}$$

*2) Recall that $f = \mu \circ h$, where $\mu$ is implemented as a single linear layer with a softmax output, and $\mathcal{L}$ is the cross-entropy loss. This combination is stictly convex with respect to the inputs to $\mu$ if they are not perfectly collinear. Therefore, by Jensen's inequality:*

$$\mathbb{E}_{\tau^{(k)}}[\mathcal{L}(f(\tau^{(k)}(\mathcal{G})), y)] > \mathcal{L}(\mu(\mathbb{E}_{\tau^{(k)}}[h(\tau^{(k)}(\mathcal{G}))]), y) \tag{20}$$

*3) This implies:*

$$\mathcal{L}_{train} > \mathbb{E}_{(\mathcal{G},y)\sim\mathcal{D}_s}[\mathcal{L}(\mu(\mathbb{E}_{\tau^{(k)}}[h(\tau^{(k)}(\mathcal{G}))]), y)], \tag{21}$$

*hence, minimizing $\mathcal{L}_{train}$ also minimizes the left hand side of the inequality.*

*4) Now, consider any target distribution $\mathcal{D}_t$ resulting from a combination of shift components in $\tau^{(k)}$. By definition, we can express $\mathcal{D}_t$ as:*

$$\mathcal{D}_t = \{\tau^{(k)}(\mathcal{G}) : \mathcal{G} \sim \mathcal{D}_s, \tau^{(k)} \sim P(\tau^{(k)})\} \tag{22}$$

*where $P(\tau^{(k)})$ is some distribution over the possible combinations of shift components.*

*5) Therefore:*

$$
\begin{aligned}
\mathbb{E}_{(\mathcal{G},y)\sim\mathcal{D}_t}[\mathcal{L}(f(\mathcal{G}), y)] &= \mathbb{E}_{(\mathcal{G},y)\sim\mathcal{D}_s}[\mathbb{E}_{\tau^{(k)}\sim P(\tau^{(k)})}[\mathcal{L}(f(\tau^{(k)}(\mathcal{G})), y)]] \\
&\leq \mathbb{E}_{(\mathcal{G},y)\sim\mathcal{D}_s}[\mathbb{E}_{\tau^{(k)}}[\mathcal{L}(f(\tau^{(k)}(\mathcal{G})), y)]] \\
&= \mathcal{L}_{train}
\end{aligned}
\tag{23}
$$

*The inequality in the second line holds because our training objective considers a wider range of transformations than those in the actual target distribution.*

*6) Equations* (21) *and* (23) *show that minimizing $\mathcal{L}_{train}$ implies both finding a model with low true risk and a model that is more invariant to $\tau^{(k)}(\mathcal{G})$, since Equation* (21) *shows the loss is lower if $\forall \tau \in supp(\tau^{(k)})$, $h(\tau(\mathcal{G})) = \mathbb{E}_{\tau^{(k)}}[h(\tau^{(k)}(\mathcal{G}))]$.*

*6) The gap between the true risk and the empirical risk can be bounded by a constant $\epsilon$ that depends on the complexity of the model and the number of samples, according to standard statistical learning theory. Therefore, we get:*

$$\mathbb{E}_{(\mathcal{G},y)\sim\mathcal{D}_t}[\mathcal{L}(f(\mathcal{G}), y)] \leq \mathbb{E}_{(\mathcal{G},y)\sim\mathcal{D}_s}[\mathbb{E}_{\tau^{(k)}}[\mathcal{L}(f(\tau^{(k)}(\mathcal{G})), y)]] + \epsilon \tag{24}$$

These theoretical results demonstrate that GraphMETRO can effectively handle complex, heterogeneous graph distribution shifts by:

- Decomposing shifts into manageable components.
- Adaptively combining expert models to handle instance-specific shifts.
- Ensuring invariance to individual and combined shift components.
- Providing a tractable upper bound on the generalization error for shifted distributions.

Compared to existing approaches that struggle with vast environment spaces or heterogeneous shifts, GraphMETRO's adaptive mixture-of-experts architecture and alignment-based training objective provide a more flexible and scalable solution for real-world graph distribution shifts.

## B    Experimental Details

**Experimental settings on synthetic datasets**. We randomly split each dataset into training (80%), validation (20%), and testing (20%) subsets. We consider transformations for $k = 2$, *i.e.,* $\tau^{(2)}$, which includes both single transformations and compositions of two different transformation functions. For the compositions, we exclude trivial combinations (e.g., adding and dropping edges) and combinations that may result in an empty graph (e.g., random subgraph sampling and node dropping). These transformations are applied to the testing datasets to create multiple variants for testing environments.

**Model architecture and optimization**. We summarize the model architecture and hyperparameters for synthetic experiments (Section 4.2) in Table 2. We use the Adam optimizer with weight decay set to 0. The encoder (backbone) architecture, including the number of layers and hidden dimensions, is selected based on validation performance from the ERM model and then fixed for each encoder during GraphMETRO training.

| | Node classification | | Graph classification | |
|---|---|---|---|---|
| | DBLP | CiteSeer | IMDB-MULTI | REDDIT-BINARY |
| Backbone | Graph Attention Networks (GAT) [63] | | | |
| Activation | PeLU | | | |
| Dropout | 0.0 | | | |
| Number of layers | 3 | 3 | 2 | 2 |
| Hidden dimension | 64 | 32 | 128 | 128 |
| Global pool | NA | NA | global add pool | global add pool |
| Epoch | 100 | 200 | 100 | 100 |
| Batch size | NA | NA | 32 | 32 |
| ERM Learning rate | 1e-3 | 1e-3 | 1e-4 | 1e-3 |
| GraphMETRO Learning rate | 1e-3 | 1e-3 | 1e-4 | 1e-3 |

Table 2: Architecture and hyperparameters on synthetic experiments.

For the real-world datasets, we use the same encoder and classifier from the implementation of the GOOD benchmark[4]. The results for the baseline methods, except for Twitter (recently added to the benchmark), are reported by the GOOD benchmark. We summarize the architecture and hyperparameters used for real-world experiments below.

| | Node classification | | Graph classification | |
|---|---|---|---|---|
| | WebKB | Twitch | Twitter | SST2 |
| Backbone | Graph Convolutional Network [25] | | Graph Isomorphism Network [70] w/ Virtual node [18] | |
| Activation | ReLU | | | |
| Dropout | 0.5 | | | |
| Number of layers | 3 | | | |
| Hidden dimension | 300 | | | |
| Global pool | NA | NA | global mean pool | global mean pool |
| Epoch | 100 | 100 | 200 | 200 |
| Batch size | NA | NA | 32 | 32 |
| ERM Learning rate | 1e-3 | 1e-3 | 1e-3 | 1e-3 |
| GraphMETRO Learning rate | 1e-2 | 1e-2 | 1e-3 | 1e-3 |

Table 3: Architecture and hyperparameters on real-world datasets.

For all datasets, we conduct a grid search for GraphMETRO learning rates due to the difference in architecture compared to traditional GNN models. GraphMETRO uses multiple GNN encoders, serving as expert modules.

## C   Stochastic Transform Functions

We built a library of 11 stochastic transform functions on top of PyG[5], and used 5 of them in our synthetic experiments for demonstration purposes. Each function allows for one or more hyperparameters to control the degree of the transformation, such as the probability parameter in a Bernoulli distribution for dropping edges. A certain amount of randomness is retained in each stochastic transform function, ensuring diversity in the generated graphs.

---

[4]https://github.com/divelab/GOOD/tree/GOODv1
[5]https://github.com/pyg-team/pytorch_geometric

```
stochastic_transform_dict = {

    'mask_edge_feat': MaskEdgeFeat(p, fill_value),
    'noisy_edge_feat': NoisyEdgeFeat(p),
    'edge_feat_shift': EdgeFeatShift(p),
    'mask_node_feat': MaskNodeFeat(p, fill_value),
    'noisy_node_feat': NoisyNodeFeat(p),
    'node_feat_shift': NodeFeatShift(p),
    'add_edge': AddEdge(p),
    'drop_edge': DropEdge(p),
    'drop_node': DropNode(p),
    'drop_path': DropPath(p),
    'random_subgraph': RandomSubgraph(k)

}
```

We observed that different sets or numbers of transform functions can impact model performance. Specifically, we use stochastic transform functions as the foundation for the decomposed target distribution shifts. Ideally, these functions should be diverse and cover different potential aspects of distribution shifts. However, using a large number of transform functions increases the demand on the gating model's expressiveness, as it must distinguish between different transformed graphs. Additionally, more transform functions increase computational cost due to the larger number of experts. An ablation study in Appendix D.3 further validates this analysis.

In practice, the stochastic transform functions proved effective on real-world datasets, suggesting their ability to represent various distribution shifts. Exploring common base transform functions to better capture real-world distribution shifts would be an interesting direction for future research.

## D Ablation Studies

### D.1 Design Choices of Expert Models

|                        | WebKB | Twitch | Twitter | SST2  |
|------------------------|-------|--------|---------|-------|
| GraphMETRO (original)  | 41.11 | 53.50  | 57.24   | 81.87 |
| GraphMETRO (w/o L1)    | 23.22 | 50.58  | 56.14   | 78.98 |
| GraphMETRO (Shared)    | 31.14 | 52.69  | 57.15   | 81.68 |

Table 4: Experiment results comparing different design choices for expert models. Results are averaged over five runs.

In the main paper, we discussed the trade-off between model expressiveness and memory utilization in expert model design. Here, we investigate a configuration where multiple experts share a GNN encoder but use individual MLPs to customize their output representations. Table 4 presents the comparative results.

Our experiments show a performance decrease when sharing the GNN encoder, which we attribute to limitations in the expressiveness of the customized modules. This may hinder alignment with the reference model and reduce the experts' ability to remain invariant to specific shift components. The concept of "being invariant to all shifts" using a shared module seems insufficient in this case. Nevertheless, this configuration still outperforms baseline models from Table 1, thanks to the gating model's ability to selectively use relevant experts and the objective function's ability to generate invariant representations.

### D.2 Alignment Design

When the alignment term is removed ($\lambda = 0$), performance drops significantly, especially for WebKB, where accuracy falls from 41.11 to 18.79. This suggests that without alignment, the expert models develop distinct representation spaces, which, when aggregated, lead to higher variance and

|                          | WebKB | Twitch | Twitter | SST2  |
|--------------------------|-------|--------|---------|-------|
| GraphMETRO (original)    | 41.11 | 53.50  | 57.24   | 81.87 |
| GraphMETRO ($\lambda = 0$) | 18.79 | 50.88  | 56.97   | 81.15 |

Table 5: Validating GraphMETRO design to align expert models with the reference model.

loss of useful information. The predictor heads, such as MLPs, struggle to process these mixed representations. The alignment mechanism is thus crucial for maintaining a coherent representation space, allowing the model to capture interactions more effectively and improving overall performance.

### D.3 Choice of Transform Functions

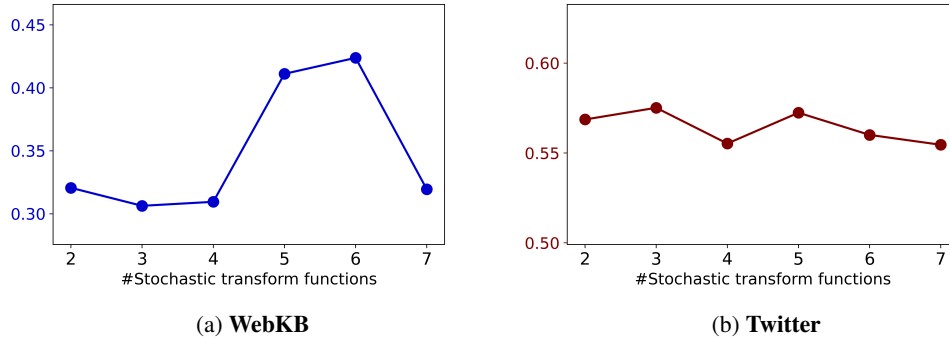

(a) **WebKB**  (b) **Twitter**

Figure 5: Impact of transform function choices on model performance. Each number of transform functions corresponds to a specific set of transformations.

We investigate how the choice and number of stochastic transform functions impact the performance of GraphMETRO , ranging from 2 to 7 functions. These functions are applied in the following sequential order:

```
[noisy_node_feat, add_edge, drop_edge, drop_node,
random_subgraph, drop_path, node_feat_shift]
```

We use the first $n$ functions and their paired combinations (excluding trivial combinations like adding and dropping edges) during the training of GraphMETRO . Due to computational constraints, we do not explore all possible combinations of the $n$ distinct functions but focus on specific sets of transformations.

Figure 5 shows the results on the WebKB and Twitter datasets. A consistent trend emerges: increasing the number of stochastic transform functions generally leads to a decline in performance. For example, performance on WebKB drops from 42.4% to 31.9%. This decline can be attributed to: (1) some stochastic functions introducing noise unrelated to the target distribution shifts, and (2) the gating model's expressiveness being insufficient to handle a larger number of transformations, leading to noisier predictions.

## E  Numerical results of Accuracy on Synthetic Distribution Shifts

Tables 6 and 7 present the numerical results on synthetic datasets corresponding to Figure 3, enabling a more detailed interpretation of the results. Additionally, we compute the average performance across different extrapolated testing datasets, showing an overall improvement.

## F  Open Discussion and Future Works

**Performance of the gating model**. The performance of GraphMETRO depends in part on how effectively the gating model can identify distribution shifts from the transform functions. Some functions,

| | DBLP | | | CiteSeer | | |
|---|---|---|---|---|---|---|
| | ERM | ERM-Aug | GraphMETRO | ERM | ERM-Aug | GraphMETRO |
| i.i.d. (0) | 85.71 | 85.66 | 85.92 | 75.80 | 76.00 | 78.01 |
| random subgraph (1) | 84.48 | 85.29 | 85.78 | 75.47 | 75.82 | 77.01 |
| drop node (2) | 71.08 | 74.85 | 76.61 | 62.21 | 63.89 | 66.22 |
| drop edge (3) | 79.69 | 82.34 | 82.95 | 71.48 | 73.24 | 77.00 |
| add edge (4) | 83.41 | 84.44 | 84.98 | 74.29 | 74.87 | 77.26 |
| noisy features (5) | 76.90 | 72.81 | 81.32 | 85.28 | 82.97 | 88.43 |
| (1, 3) | 77.63 | 81.04 | 81.71 | 70.37 | 71.42 | 74.97 |
| (2, 3) | 81.99 | 83.65 | 84.26 | 73.60 | 74.06 | 76.11 |
| (1, 4) | 79.69 | 68.62 | 80.31 | 84.47 | 86.36 | 88.56 |
| (2, 4) | 70.55 | 74.01 | 75.10 | 62.13 | 63.53 | 65.73 |
| (1, 5) | 71.52 | 68.27 | 71.05 | 66.89 | 62.59 | 67.32 |
| (2, 5) | 77.73 | 81.13 | 81.85 | 70.19 | 72.21 | 76.77 |
| (3, 5) | 79.59 | 84.49 | 87.14 | 78.24 | 73.29 | 89.18 |
| (4, 5) | 70.40 | 74.16 | 76.18 | 61.64 | 63.53 | 66.42 |
| Average | 77.88 | 78.63 | 81.08 | 72.29 | 72.41 | 76.36 |

Table 6: Numerical results on synthetic node classification datasets

| | IMDB-MULTI | | | REDDIT-BINARY | | |
|---|---|---|---|---|---|---|
| | ERM | ERM-Aug | GraphMETRO | ERM | ERM-Aug | GraphMETRO |
| i.i.d. (0) | 50.17 | 49.28 | 49.16 | 72.93 | 73.02 | 75.94 |
| random subgraph (1) | 34.30 | 39.94 | 45.86 | 62.59 | 69.03 | 71.22 |
| drop node (2) | 50.42 | 48.73 | 48.83 | 70.01 | 72.27 | 72.26 |
| drop edge (3) | 49.66 | 48.94 | 48.83 | 59.13 | 70.55 | 72.51 |
| add edge (4) | 49.64 | 48.14 | 48.90 | 65.18 | 67.28 | 69.34 |
| noisy features (5) | 50.17 | 49.28 | 49.16 | 68.66 | 68.50 | 66.79 |
| (2, 3) | 34.55 | 40.32 | 45.11 | 58.72 | 64.06 | 66.50 |
| (1, 4) | 34.32 | 40.28 | 46.01 | 59.40 | 62.81 | 65.29 |
| (2, 4) | 34.57 | 40.17 | 46.79 | 61.34 | 66.02 | 66.71 |
| (1, 5) | 49.31 | 48.36 | 48.68 | 65.89 | 66.88 | 68.09 |
| (2, 5) | 50.51 | 48.78 | 48.79 | 68.72 | 69.77 | 68.76 |
| (3, 5) | 49.38 | 47.72 | 48.35 | 55.36 | 65.21 | 64.87 |
| (1, 3) | 48.72 | 48.36 | 48.76 | 61.08 | 61.71 | 62.57 |
| (4, 5) | 34.62 | 39.88 | 46.15 | 62.99 | 68.68 | 68.34 |
| Average | 44.31 | 45.58 | 47.82 | 63.71 | 67.56 | 68.51 |

Table 7: Numerical results on synthetic graph classification datasets

like adding node feature noise and extracting random subgraphs, are inherently disentangled, making it easy for the gating model to differentiate between these distributions. Other functions, such as dropping paths and dropping edges, may be more similar, but the method remains robust as long as each expert produces the corresponding invariant representation. More complex combinations of transforms pose a greater challenge for the gating model's expressiveness. To address this, initializing the gating model with a pre-trained model from a diverse dataset may enhance its ability to predict mixtures, improving performance on unseen graphs.

**Comparison with invariant learning methods**. GraphMETRO differs from traditional invariant learning, where environments are constructed using environment variables. Instead, GraphMETRO views distribution shifts on an instance as a mixture, represented by the score vector from the gating function. This approach enables the creation of infinite environments, as the score vector is continuous. When restricting the gating function to binary outputs, GraphMETRO can simulate finite environments, akin to the environment construction in invariant learning. Additionally, the concept of referential invariant representation using the base model $\xi_0$ sets GraphMETRO apart from previous invariant learning approaches.

**Applicability of GraphMETRO** . A key question is how well the predefined transform functions capture complex distribution shifts.

- **General domain:** In our experiments, we primarily use five universal graph augmentations (as listed in [82]). Our code also includes additional transforms (Appendix C). While these transforms are not exhaustive, they cover a wide range of shifts observed in our results. However, real-world

distribution shifts may go beyond the predefined transforms, and in such cases, GraphMETRO might struggle to capture and mitigate unknown shifts. This is a limitation when the test distribution or domain knowledge is insufficient.

- **Specific domains:** In certain domains, additional knowledge can help infer distribution shifts, such as an increase in malicious users in a trading system. This knowledge can guide the construction of transform functions to better cover the target distribution shifts. Specifically, two sources of knowledge can be used: **i) Domain knowledge**, *e.g.,* in molecular datasets, transform functions could add carbon structures to molecules while preserving functional groups, or in social networks, known user behaviors can guide transformations. **ii) Leveraging samples from the target distribution (*i.e.,* domain adaptation)**, where samples from the target can inform the selection of relevant transforms. For example, by measuring the distance between the extrapolated datasets under specific transforms and the target samples in the embedding space, more relevant transform functions can be selected. This presents an interesting direction for future work.

**Label distribution shifts**. In this work, we focus on distribution shifts in graph structures and features. Extending GraphMETRO to handle label distribution shifts would be a complementary and interesting direction. Label shifts affect various modalities, including graphs and images, and existing methods [44, 5] designed for label shifts could be integrated into our framework with minimal adjustments, such as modifying the loss function or training pipeline.

