# OpenReview forum: "GraphMETRO: Mitigating Complex Graph Distribution Shifts via Mixture of Aligned Experts"
_NeurIPS.cc/2024/Conference — NeurIPS 2024 poster_

### Official Review · Reviewer_ymW6 · 2024-07-12

**Soundness:** 3
**Presentation:** 3
**Contribution:** 3
**Rating:** 5
**Confidence:** 4

**Summary:**

The manuscript presents a model for handling complex distribution shifts in graph data. The proposed GraphMETRO model employs a mixture-of-experts (MoE) architecture, with a gating model to identify distributional shifts and multiple expert models to generate shift-invariant representations. This method aims to generalize graph neural networks (GNNs) to non-synthetic distribution shifts occurring naturally in real-world graph data, achieving state-of-the-art results on several datasets.

**Strengths:**

1. This paper studies an interesting research problem that is handling complex distribution shifts in graph data. This research problem has been very hot recently.
2. The model design is easy to understand. The paper provides a detailed explanation of the proposed model.
3. The experiment demonstrates the effectiveness of the model. The performance improvement on some comparisons seems to be significant.

**Weaknesses:**

1. Although the motivation is clear to me, the novelty is one of my concerns since explicitly considering instance heterogeneity for graph OOD problems is not new. I think the authors could address my concern by presenting more differences with related works clearly.
2. There are no detailed theoretical analyses to demonstrate why the model can well address the problem and perform better than the baselines.
3. The reviews of the related works are not enough. Please refer to the paper Out-Of-Distribution Generalization on Graphs: A Survey for a more comprehensive discussion of the related works on graph OOD problems. For example, the discussions on invariant learning in section 2 are too brief. The main claim “the standard invariant learning approaches are not well-equipped to mitigate the complex distribution shifts” is a little arbitrary considering the graph OOD literature.

**Questions:**

Could you provide detailed theoretical analyses to demonstrate why the model can excellently address the problem and perform better than the baselines?

**Limitations:**

Yes

---

> ### Author Rebuttal · Authors · 2024-08-07
>
> We are grateful for your detailed suggestions! We provide responses below:
>
> ## **Comment 1,3: A thorough review on related works and highlights on the novelty**
>
> Thanks for the comment! We added more related works, including those in [2]:
>
> - **Added Related work**: Recent methods for improving OOD generalization in GNNs include OOD-GNN, which uses nonlinear decorrelation to eliminate spurious correlations [3]; OOD-GMixup, which creates virtual OOD samples by perturbing graph rationale space [5]; and OODGAT, which models interactions between in-distribution and OOD nodes [6]. G-Mixup generates synthetic graphs by interpolating graphons [1], while AIA uses adversarial augmentation to create new environments while preserving stable features [7]. GREA employs environment replacement for better graph rationalization [4], and DPS constructs diverse populations from source domains to train an equi-predictive GNN [8].
> - **Novelty of GraphMETRO with comparison**: GraphMETRO differs fundamentally by decomposing complex distributional shifts into interpretable shift components, enabling it to mitigate a combinatorial number of potential shift variations. While previous works [1,3,4,5] consider heterogeneity, GraphMETRO offers a more fine-grained level of heterogeneity by modeling a continuous space of environments, handling infinite potential shift scenarios. Its concept of referential invariant representation and expert model alignment aid generating robust representations.
>
> Please let us know if this clears the concern! Due to the character limit, we are happy to provide more detailed comparisons with individual works.
>
> ## **Comment 2: Detailed theoretical analysis why GraphMETRO addresses the problem and outperforms baselines.**
>
> Thank you for the opportunity to clarify the theory. Here is part of the updated Appendix A.
>
> ---
>
> ### **Theoretical Guarantees**
> **Theorem 1 (Shift-Invariance)** For any graph $\mathcal{G}$ and shift component $\tau_i$, the encoder $h$ satisfies: $h(\tau_i(\mathcal{G})) = h(\mathcal{G})$ ...
>
> **Theorem 2 (Composition of Shifts)** For any graph $\mathcal{G}$ and combination of $k$ shift components $\tau^{(k)} = \tau_{i_1} \circ \tau_{i_2} \circ ... \circ \tau_{i_k}$, the encoder $h$ approximately satisfies: $h(\tau^{(k)}(\mathcal{G})) \approx h(\mathcal{G})$ ...
>
> **Theorem 3 (Generalization Bound)** Let $\mathcal{L}(\cdot, \cdot)$ be the cross-entropy loss. For any distribution $\mathcal{D}_t$ resulting from a combination of shift components in $\tau^{(k)}$, the generalization error of GraphMETRO satisfies:
>
> $$
> \mathbb{E} _{(\mathcal{G}, y) \sim \mathcal{D} _t} [\mathcal{L}(f(\mathcal{G}), y)] \leq \mathbb{E} _{(\mathcal{G}, y) \sim \mathcal{D} _s} [\mathbb{E} _{\tau^{(k)}} [\mathcal{L}(f(\tau^{(k)}(\mathcal{G})), y)]] + \epsilon
> $$
>
> where $\epsilon$ is a small constant depending on the complexity of the model and the number of samples.
>
> **Proof:**
>
> 1) Our training objective minimizes:
>
> $$
> \mathcal{L} _\text{train} = \mathbb{E} _{(\mathcal{G}, y) \sim \mathcal{D} _s} [\mathbb{E} _{\tau^{(k)}} [\mathcal{L}(f(\tau^{(k)}(\mathcal{G})), y)]]
> $$
>
> 2) Recall that $f = \mu \circ h$, where $\mu$ is implemented as a single linear layer with a softmax output, and $\mathcal{L}$ is the cross-entropy loss. This combination is strictly convex with respect to the inputs to $\mu$ if they are not perfectly collinear. Therefore, by Jensen's inequality:
>
> $$
> \mathbb{E} _{\tau^{(k)}} [\mathcal{L}(f(\tau^{(k)}(\mathcal{G})), y)] > \mathcal{L}(\mu(\mathbb{E} _{\tau^{(k)}}[h(\tau^{(k)}(\mathcal{G}))]), y)
> $$
>
> 3) This implies:
>
> $$
> \mathcal{L} _\text{train} > \mathbb{E} _{ (\mathcal{G}, y) \sim \mathcal{D} _s} [\mathcal{L}(\mu(\mathbb{E} _{\tau^{(k)}}[h(\tau^{(k)}(\mathcal{G}))]), y)],
> $$
>
> hence, minimizing $\mathcal{L}_\text{train}$ also minimizes the right-hand side of the inequality.
>
> 4) Consider any target distribution $\mathcal{D}_t$ resulting from a combination of shift components in $\tau^{(k)}$. By definition, we can express $\mathcal{D}_t$ as a set of $\tau^{(k)}(\mathcal{G})$, where $\mathcal{G} \sim \mathcal{D} _s, \tau^{(k)} \sim P(\tau^{(k)})$. Here $P(\tau^{(k)})$ is some distribution over possible combinations of shift components.
>
> 5) Therefore:
>
> $$
> \mathbb{E} _{(\mathcal{G}, y) \sim \mathcal{D} _t} [\mathcal{L}(f(\mathcal{G}), y)] = \mathbb{E} _{(\mathcal{G}, y) \sim \mathcal{D} _s} [\mathbb{E} _{\tau^{(k)} \sim P(\tau^{(k)})} [\mathcal{L}(f(\tau^{(k)}(\mathcal{G})), y)]]
> \leq \mathbb{E} _{(\mathcal{G}, y) \sim \mathcal{D} _s} [\mathbb{E} _{\tau^{(k)}} [\mathcal{L}(f(\tau^{(k)}(\mathcal{G})), y)]]
> = \mathcal{L} _\text{train}
> $$
>
> The inequality here holds because our training objective considers a wider range of transformations than those in the actual target distribution.
>
> 6) Equations in steps (3) and (5) show that minimizing $\mathcal{L} _\text{train}$ implies finding a model with low true risk and one more invariant to $\tau^{(k)}(\mathcal{G})$, since step (3) shows the loss is lower if $\forall \tau \in \text{supp}(\tau^{(k)})$, $h(\tau(\mathcal{G})) = \mathbb{E} _{\tau^{(k)}}[h(\tau^{(k)}(\mathcal{G}))]$.
>
> 7) The gap between the true risk and the empirical risk can be bounded by a constant $\epsilon$. Therefore, we get:
>
> $$
> \mathbb{E} _{(\mathcal{G}, y) \sim \mathcal{D} _t} [\mathcal{L}(f(\mathcal{G}), y)] \leq \mathbb{E} _{(\mathcal{G}, y) \sim \mathcal{D} _s} [\mathbb{E} _{\tau^{(k)}} [\mathcal{L}(f(\tau^{(k)}(\mathcal{G})), y)]] + \epsilon
> $$
>
> This gives GraphMETRO advantages over baselines, especially with heterogeneous environments, which generally assume the environment space $\mathbb{E}$ in $P(Y | f(\mathcal{G}), e\in\mathbb{E}) \approx P(Y | f(\mathcal{G}))$ involves relatively simple components.
>
> ## **Summary**
>
> We thank you for your approval on our motivation, presentation, and effectiveness. We appreciate it if you could reconsider your evaluation if some concerns on novelty and theoretical analysis are addressed. Thanks!
>
> Reference: Please see the PDF.

---

> > ### Comment · Reviewer_ymW6 · 2024-08-13
> >
> > Thank the authors for the detailed responses. I will increase my score.
> >
> > A minor suggestion for the revised paper is to keep the newly added 8 references in the General Response PDF in the same format as the references in the original paper. I noticed that the formatting of references doesn't seem to be standardized, with abbreviations and full names being mixed up.

---

> > > ### Author Response · Authors · 2024-08-13
> > > **Thanks!**
> > >
> > > We thank reviewer ymW6 for the approval! We will make sure all references show full names appropriately.

---

### Official Review · Reviewer_t31K · 2024-07-12

**Soundness:** 2
**Presentation:** 3
**Contribution:** 2
**Rating:** 6
**Confidence:** 4

**Summary:**

This paper introduces a novel Graph Neural Network (GNN) architecture designed to handle complex distribution shifts in graph data. GraphMETRO uses a Mixture-of-Experts (MoE) model, with multiple expert models each targeting specific distribution shifts. Additionally, this paper uses a gating model to identify the most portable shifts. The approach aims to produce invariant representations, aligning expert outputs for smooth optimization. GraphMETRO demonstrates state-of-the-art results on various datasets, outperforming existing methods in generalizing to real-world distribution shifts.

**Strengths:**

1.	This paper proposes a framework that is able to decompose any distribution shift into multiple shift components.

2.	Based on the MoE structure, the authors design an architecture with a gating model and multiple expert models to identify strong distribution shift components.

3.	The proposed method achieves state-of-the-art results on real-world GOOD benchmarks and provides interpretability of distribution shifts on unknown datasets

**Weaknesses:**

1.	The alignment of expert representations with the reference model using the Frobenius norm might lead to the incorporation of undesired information (e.g., node degree or feature noise). The impact of this on the generalizability of the proposed framework remains unclear.

2.	Although each expert model is intended to correspond to an individual shift component, the outputs of a GNN or MLP may include extraneous information beyond the distribution shift [1]. This raises questions about whether the expert model can consistently produce invariant features.

3.	The number of experts is determined by the parameter k, which influences the identification of individual shift components. The choice of k could impact the generalizability of the framework, yet the effects of different k values on the framework’s performance are not clearly elucidated.

[1] Li, Haoyang, et al. "Disentangled Graph Self-supervised Learning for Out-of-Distribution Generalization." Forty-first International Conference on Machine Learning. 2024

**Questions:**

Based on the weakness section, the questions are as follows:

1.	Does each expert encode undesired information from the reference model?
2.	Can the expert model consistently produce invariant features using GNN or MLP architectures?
3.	How does the choice of the parameter k impact the model’s generalizability?

**Limitations:**

The authors adequately addressed the limitations and potential negative societal impact of the work.

---

> ### Author Rebuttal · Authors · 2024-08-06
>
> We appreciate your efforts and insightful comments! To address your concerns, we provide detailed responses below.
>
> ---
>
> ## **Comment 1: Does each expert encode undesired information from the reference model?**
>
> Great catch! Previously we conducted experiments to explore the impact of aligning each expert with the reference model. We found that results without alignment were poor, with final representations appearing mixed which we observed by t-SNE visualization.
>
> To formally validate this observation, here are the results (repeated three times for $\lambda=0$):
>
> |                    | WebKB | Twitch | Twitter | SST2  |
> |--------------------|-------|--------|---------|-------|
> | $\lambda=0$        | 18.79 | 50.88  | 56.97   | 81.15 |
> | $\lambda=1$ (original results) | 41.11 | 53.50  | 57.24   | 81.87 |
>
> In our paper, we had attributed this effect to two reasons:
>
> - Each expert model may develop its own unique representation space, leading to information loss when aggregated with other expert outputs.
> - Aggregating independent representations creates a mixed space with high variance, which is difficult for predictor heads like MLPs to process and capture interactions among diverse representations.
>
> **Therefore, training without alignment poses challenges.** However, the reviewer raised an interesting point about the risks of learning undesirable biases from the reference model. We believe **the risk can be mitigated** by preventing the reference model from overfitting on training data, thereby reducing bias during the training. We have added this ablation result to the Appendix and included this discussion in our methodology section.
>
> ---
>
> ## **Comment 2: Can the expert model consistently produce invariant features using GNN or MLP architectures?**
>
> Great question! The short answer is **yes**. Please see Figure 4(a) and Section 4.3, where we compute an invariant matrix showing that the representation produced by the expert closely matches that of the reference model. This indicates that the expert model consistently produces (approximately) invariant features with respect to their shift components.
>
> Please let us know if this answers your question; we are happy to provide further validation if needed.
>
> ---
>
> ## **Comment 3: How does the choice of the parameter $k$ impact the model’s generalizability?**
>
> Thanks for asking! In `Appendix E: Study on the Choice of Transform Functions`, we have explored varying numbers of transform functions, i.e., $k$.
>
> Due to character limits, we can't include the full content here. In summary, when $k$ is too large, performance may decline due to noise or "confusion" for the gating model. Please see the detailed analysis in Appendix E.
>
> We also would like to mention that our framework involves a relatively small number of extra hyperparameters, including $k$ and $\lambda$ in the training objective, compared to training a standard GNN. We hope this study provides more insight into the mechanism of our framework.
>
> ---
>
> ## **Summary**
>
> We hope to address your concerns with:
>
> 1. Our additional ablation study and discussion on expert alignment.
> 2. Reference to the study on whether expert models consistently produce invariant features.
> 3. Reference to the study on the parameter $k$.
>
> We also appreciate your support and positive feedback on our presentation, novelty, and effectiveness. We hope our response justifies the properties of GraphMETRO training, leading to improved soundness and interpretability. We would greatly appreciate your reconsideration of the empirical soundness of our work given our response.
>
> Thank you again!

---

> > ### Comment · Reviewer_t31K · 2024-08-12
> >
> > The authors addressed my concern with additional experiments and an ablation study, so I remain my positive score 6.

---

### Official Review · Reviewer_LHfk · 2024-07-15

**Soundness:** 3
**Presentation:** 3
**Contribution:** 2
**Rating:** 5
**Confidence:** 3

**Summary:**

This paper proposed GraphMETRO, a novel Graph Neural Network (GNN) architecture that try to models complex and natural distribution shifts in real-world graph data. The key innovation is a mixture-of-experts (MoE) approach, where the model decomposes the distribution shift into multiple shift components, each handled by a dedicated expert model. The proposed  GraphMETRO is expected to identify the shift type for a given input graph, and  generate representations that are invariant to the assigned shift components. The experiments show that GraphMETRO can capture heterogeneous shifts across instances and outperform baseline methods on diverse real-world datasets exhibiting different distribution shifts.

**Strengths:**

•	This paper proposes a novel method to addresses distribution shifts in real-world graph data from a MOE perspective.

•	This paper is well-written and it provides a detailed experimental analysis.

•	This empirical validation demonstrates the effectiveness and applicability of the proposed method.

**Weaknesses:**

•	The paper proposes five types of graph distribution shifts (adding edges, deleting edges, feature noise, subgraphs, deleting nodes), but real-world graph distribution shifts may be more varied, such as systematic changes in link preferences and the addition of malicious nodes.

•	The comparison methods in the experiment lack the latest OOD methods such as OOD-GNN, OOD-GAT-ATT, and OOD-GMixup and suggested to supplement the experimental results on more datasets.

•	The alignment design of different expert models lacks experimental support, and it is recommended to supplement with corresponding ablation experiments.

**Questions:**

•	Why distributional difference can be decomposed into the proposed five shifts? How to deal with complex distribution shifts in the real world that are out of the scope of these five shifts?

•	Should the expert model and Gating be trained separately or together, and why was it designed this way?

**Limitations:**

please refer to the weakness

---

> ### Author Rebuttal · Authors · 2024-08-07
>
> We thank the reviewer for the insightful comments. Please find our responses to each point below.
>
> ---
>
> ## **Comment 1: Real-world graph shifts may be more varied than defined shifts**
>
> Great question! We discussed this in `Appendix G: Open Discussion and Future Works`, focusing on `The applicability of GraphMETRO` under complex scenarios and potential mitigations:
> - **Specific Domains:** Real-world graphs can vary significantly, but domain knowledge and observations help address this issue. Insights into distribution shifts, such as the rise of malicious users in `trading systems`, aid in creating transform functions that cover target shifts. This knowledge can come from:
>   - **Domain Knowledge:** In `molecular datasets`, transform functions might add carbon structures while preserving functional groups. In `social networks`, they can be based on known user behaviors.
>   - **Domain Adaptation:** Using a few samples from the target distribution can guide the selection or creation of transform functions. This ensures coverage of distribution shifts. For example, selecting transform functions by measuring the distance between extrapolated datasets and target samples in the embedding space is a promising future direction.
>
> - **General Domains:** Our experiments utilize five stochastic transform functions, universal graph augmentations as listed in Zhao et al. (2021). While not exhaustive, these functions effectively cover a wide range of distribution shifts, supported by our results. We provide a **theoretical guarantee** (see our response to `reviewer ymW6`) that bounds the generalization error on testing data under the assumption of shared support between testing and training transformations.
>
> - **Further Considerations:** Although our approach is robust, real-world graph distribution shifts can be highly complex, and even mild assumptions may not always hold. Such scenarios occur due to limited knowledge of the testing distribution, which is a common challenge for out-of-distribution generalization methods. We see this as an opportunity for future research to **proactively capture and anticipate distribution shift types.**
>
>
> We appreciate your insights! We hope this discussion addresses your concerns.
>
> ---
>
> ## **Comment 2: Adding latest OOD methods such as OOD-GNN, OODGAT-ATT, and OOD-GMixup**
>
> Thanks for bringing up these relevant works! We included two more baselines: OODGAT [1] (for node classification only) and G-Mixup [4] (for graph classification only), though we couldn't find existing code for OOD-GNN [2] and OOD-GMixup [3]. Please let us know if otherwise.
>
> || WebKB     | Twitch| Twitter    | SST2       |
> |:--|:--|:--|:--|:--|
> | OODGAT| 14.41±1.10| 49.38±0.87 | -| -|
> | G-Mixup| -| -| 53.32±2.75 | 77.43±1.97 |
> | GraphMETRO| 41.11±7.47| 53.50±2.42 | 57.24±2.56 | 81.87±0.22 |
>
> The results for OODGAT and G-Mixup are repeated three times. For G-Mixup, we used the same GNN architectures and common hyperparameters (e.g., lr) specified in Appendix B. For OODGAT, due to its unique architecture design, we used their original implementation including hyperparameters (e.g., $a=0.9, b=0.01$).
>
> We found that OODGAT performs well on the Twitch dataset. However, G-Mixup struggled to generate effective synthetic node features for semantic graphs, particularly when node dimensions are large (768 on both Twitter and SST2).
>
> We have added more discussions about these three works to the related work. Please see our response to `reviewer ymW6` if you are interested!
>
> ---
>
> ## **Comment 3: Experimental support of the alignment design of different expert models**
>
> Great catch! We previously conducted experiments to explore the impact of aligning each expert with the reference model. We found that results without alignment were poor, with final representations appearing mixed, as observed through t-SNE visualization.
>
> To formally validate this observation, here are the results (repeated three times for $\lambda=0$):
>
> |                    | WebKB | Twitch | Twitter | SST2  |
> |--------------------|-------|--------|---------|-------|
> | $\lambda=0$        | 18.79 | 50.88  | 56.97   | 81.15 |
> | $\lambda=1$ (original results) | 41.11 | 53.50  | 57.24   | 81.87 |
>
> In our paper, we attributed this effect to two reasons:
>
> - Each expert model may develop its own unique representation space, leading to information loss when aggregated with other expert outputs.
> - Aggregating independent representations creates a mixed space with high variance, which is difficult for predictor heads like MLPs to process and capture interactions among diverse representations.
>
> **Therefore, training without alignment poses challenges.** We hope this validates our alignment design.
>
> ---
>
> ## **Summary**
>
> We are grateful for your positive evaluation of our presentation, novelty, and effectiveness. We further provide the following responses:
>
> 1. An in-depth discussion about applicability.
> 2. Additional experiments on baselines.
> 3. An additional ablation study on the alignment impact.
>
> We hope our responses address your concerns. We would like to highlight that our main contribution is framing the graph OOD problem into a simple yet novel formulation that is more tractable, as well as proposing the MoE training framework with theoretical guarantees.
>
> Overall, we believe our work makes good contributions to the field of graph distribution learning, and we would appreciate your reconsideration. Thank you for your efforts again!
>
> ---
>
> [1] Song et al. Learning on Graphs with Out-of-Distribution Nodes. KDD 2022.
>
> [2] Li et al. OOD-GNN: Out-of-Distribution Generalized Graph Neural Network.
>
> [3] Lu et al. Graph Out-of-Distribution Generalization with Controllable Data Augmentation.
>
> [4] Han et al. G-Mixup: Graph Data Augmentation for Graph Classification. ICML 2022.

---

### Official Review · Reviewer_uY6m · 2024-07-15

**Soundness:** 3
**Presentation:** 3
**Contribution:** 3
**Rating:** 7
**Confidence:** 3

**Summary:**

This paper proposes a novel model, GraphMETRO, that applies mixture of experts (MoE) to facilitate generalizable graph neural network (GNN). It uses different expert networks to mitigate different types of distribution shifts, such as adding edges, dropping nodes, noisy features, etc. When facing a distribution shift, the paper assumes that the distribution shift can be decomposed as a combination of different types of distribution shifts. Each expert network can generate invariant embeddings under a specific type of distribution shift. A gating network is in charge of deciding the types of distribution shifts. Therefore, GraphMETRO can automatically identify the types of distribution shifts and generate invariant embeddings.

**Strengths:**

(1) The studied problem is interesting and worth studying, and the motivation is meaningful. MoE has been used in domain transfer and domain generalization in vision and NLP tasks, but using it in GNN to facilitate distribution shift is novel and worth studying.

(2) The model design is novel and reasonable. It also has good interpretability.

(3) Extensive experiments verify the effectiveness of GraphMETRO.

**Weaknesses:**

(1)   The actual distribution shift can be very complex, but the paper only considers a few basic types of distribution shifts. They might not be able to represent some distribution shifts in the real world.

(2)   Some writing notations are not strict enough and can be improved. For example, in Definition 1 (referential invariant representation), in line 196, it says $\epsilon_0(G) = \epsilon^*(\tau(G))$, maybe it should be $\approx$ rather than $=$. Also, in line 218, it might be better to use $\approx$ instead of $=$.

**Questions:**

(1)   I am wondering how you obtain the ground truth labels of the types of distribution shift on the test set. I understand that for the training set, you can do data augmentation by adding different types of distribution shifts, so you have the ground truth, but for the test set, I am wondering about it.

(2)   How did you train the baseline generalization/robustness models? How did you ensure a fair comparison between GraphMETRO and the baseline methods?

**Limitations:**

The authors have discussed the limitations in the appendix.

---

> ### Author Rebuttal · Authors · 2024-08-07
>
> We are grateful for your positive feedback and detailed suggestions! We provide responses below to address your remaining concerns.
>
> ## **Comment 1: The considered distribution shifts may not fully represent real-world shifts.**
>
> Thanks for the comment! We discussed this in `Appendix G: Open Discussion and Future Works`, focusing on `The applicability of GraphMETRO` under complex scenarios and potential mitigations:
> - **Specific Domains:** Real-world graphs can vary significantly, but domain knowledge and observations help address this issue. Insights into distribution shifts, such as the rise of malicious users in `trading systems`, aid in creating transform functions that cover target shifts. This knowledge can come from:
>   - **Domain Knowledge:** In `molecular datasets`, transform functions might add carbon structures while preserving functional groups. In `social networks`, they can be based on known user behaviors.
>   - **Domain Adaptation:** Using a few samples from the target distribution can guide the selection or creation of transform functions. This ensures coverage of distribution shifts. For example, selecting transform functions by measuring the distance between extrapolated datasets and target samples in the embedding space is a promising future direction.
>
> - **General Domains:** Our experiments utilize five stochastic transform functions, universal graph augmentations as listed in Zhao et al. (2021). While not exhaustive, these functions effectively cover a wide range of distribution shifts, supported by our results. We provide a **theoretical guarantee** (see our response to `reviewer ymW6`) that bounds the generalization error on testing data under the assumption of shared support between testing and training transformations.
>
> - **Further Considerations:** Although our approach is robust, real-world graph distribution shifts can be highly complex, and even mild assumptions may not always hold. Such scenarios occur due to limited knowledge of the testing distribution, which is a common challenge for out-of-distribution generalization methods. We see this as an opportunity for future research to **proactively capture and anticipate distribution shift types.**
>
> We appreciate your insights! We hope this discussion addresses your concerns.
>
> ---
>
> ### **Comment 2: Notations**
> Thanks! We corrected the mentioned notations and rigorously checked other notations, including those used in our response to `reviewer ymW6` on the detailed theoretical analysis.
>
> ---
>
> ### **Question 1: How to obtain the ground truth labels of the types of distribution shift on the test set?**
>
> Good question! In `Figure 3: Accuracy on synthetic distribution shifts`, the testing environments on synthetic datasets are also constructed by augmentation. The difference is that testing environments could be augmented by multiple transform functions, resulting in testing graphs that are unseen during training. For real-world graphs, the ground truth distribution shifts are not available; therefore, we conducted qualitative analysis in Section 4.4.
>
> ---
>
> ### **Question 2: How did you train the baseline generalization/robustness models? How did you ensure a fair comparison between GraphMETRO and the baseline methods?**
>
> Since we are using real datasets from the GOOD benchmark, which provides implementation of the OOD baseline methods and base GNN models, we aligned our settings (Appendix C, Table 3) with them.
>
> Specifically, We adopted the same encoder and classifier from their implementation. Results of the baseline methods, except for the Twitter dataset, are officially reported by the GOOD benchmark. The results on the Twitter dataset are not available in the official leaderboard but were also produced using their code base.
>
> Therefore, we ensured the **dataset settings (splits, node features)** and **architecture components (encoder, classifier, pooling layer, etc.)** were the same to ensure a fair comparison.
>
> ---
>
> ## **Summary**
>
> We want to thank you for your insightful questions and approval of our motivation, novelty, and effectiveness.
>
> - For your applicability concern, we conducted more discussion to make it clear.
> - For notations, we hope we have resolved this issue with more rigorous checks.
>
> We sincerely hope all of your concerns are addressed. We appreciate your support and we are happy to engage further if there are any other points we missed! Thank you!

---

> > ### Comment · Reviewer_uY6m · 2024-08-14
> >
> > Thank you for your reply! Your response has resolved my concerns about the generalizability of the method. Since you can use domain knowledge and observations to define the types of distribution shifts of a specific application, and also due to the theoretical guarantee, it seems that GraphMETRO might be useful for various potential applications. I would like to raise my score to 7.

---

> > > ### Author Response · Authors · 2024-08-14
> > >
> > > We thank Reviewer uY6m for the positive consent and for providing a clear summary of our work's applicability!

---

### Official Review · Reviewer_U8um · 2024-07-18

**Soundness:** 2
**Presentation:** 3
**Contribution:** 3
**Rating:** 5
**Confidence:** 3

**Summary:**

The study investigates how to use mixture-of-experts (MoE) to model complex distribution shifts. It explores leveraging various expert modules, each tailored to different distribution shifts, to generate embeddings that enhance OOD generalization on graphs.

**Strengths:**

1. This work proposes an novel idea to make use of representations from various distribution shifts for OOD generalization, which is different from previous invariant learning and distribution robustness optimization (DRO) literatures.

2. The proposed framework can be applied for both node and graph-level classification.

3. GraphMetro can achieve significant performance improvement in some datasets with complex distribution shifts, such as WebKB.

**Weaknesses:**

1. The terminology "invariant" in this work appears to be inconsistent with its usage in the invariant learning literature. In invariant learning, "invariant" refers to the features that perform stably across different environments. However, in this work, some claims regarding "invariant" seem misleading. For instance, the statement: "we train the expert models to generate invariant representations with respect to their corresponding shift components" (lines 69-70) is unclear to me. It is not clear how the expert model can generate invariant representations given the training objective in Equation 3. Different experts are optimized using empirical risk minimization (ERM) with specific data augmentations, and there seems to be no regularization to enforce the model to learn invariant features.


2. In my opinion, GraphMetro actually utilizes spurious (style) features in different distribution shifts learned by the expert models, rather than invariant features as discussed in the paper. When the test dataset exhibits specific distribution shifts that align with the data augmentations in the training set, the gating model can select the representations learned by a subset of experts and make use of them.  A similar work that leverages spurious features for OOD generalization is [1], where the distribution shifts are reflected in different environments. Please correct me if I am wrong, further discussions regarding this are also welcomed.


3. One of my concern of this method is that the distribution shifts are correlated with random data augmentation, which may restrict its efficacy. For instance, consider scaffold shift or assay shift, which are prevalent in molecular[2] or drug-discovery[3] datasets. It is unclear whether GraphMetro would be effective for such datasets, where there is only one type of distribution shift, which may not align with a specific type of random data augmentation.


__Reference__

1. Yao et al., Improving Domain Generalization with Domain Relations, ICLR 2024.
2. Hu et al., Open Graph Benchmark: Datasets for Machine Learning on Graphs, NeurIPS 2020.
3. Ji et al., DrugOOD: Out-of-Distribution Dataset Curator and Benchmark for AI-Aided Drug Discovery – a Focus on Affinity Prediction Problems with Noise Annotations, AAAI 2023.

**Questions:**

1. How does GraphMetro perform on other datasets, such as OGBG-Molbace, OGBG-Molbbbp, or DrugOOD, which exhibit typical and single distribution shifts like scaffold shift?

---

> ### Author Rebuttal · Authors · 2024-08-07
>
> We appreciate your comments! We believe the major concern might be due to a bit of misunderstanding on the referential invariant representations and our objective. Please feel free to correct us if otherwise! Here we provide justification from a theoretical perspective for simplicity.
>
> ---
>
> ## **Comment 1, 2: GraphMETRO's Invariant Representations**
>
> ### **1) Definition of invariant representations**
>
> We define an invariant representation with respect to a stochastic transform function $\tau$ as:
>
> $\xi^*(\mathcal{G}) = \xi^*(\tau(\mathcal{G}))$
>
> This aligns with the concept of stability across different environments, where each $\tau$ creates a different environment.
>
> ### **2) Why each expert is invariant w.r.t. a shift component**
> Each expert $\xi_i$ produces **referential invariant representations (c.f. definition 1)** with respect to its shift component $\tau_i$:
>
> $\xi_0(\mathcal{G}) = \xi_i(\tau_i(\mathcal{G}))$
>
> This ensures stability under its transformation, aligned with the reference model $\xi_0$. The necessity of referential invariant representation is justified in our response to `reviewer t31K`.
>
> **Even with a single expert, the model's representation remains invariant,** as guaranteed by our Shift-Invariance Theorem, explained in point 4 later.
>
> ### **3) Expert model invariance w.r.t. a shift component given the objective**
>
> The distance term in Equation 3:
>
> $d(h(\tau^{(k)}(\mathcal{G})), \xi_0(\mathcal{G}))$
>
> encourages each expert to align its output with the reference model $\xi_0$. This, combined with the gating model, leads to invariant representations by MoE.
>
> ### **4) How the MoE model (gating + experts) generates invariant representations**
>
> **Theorem 1 (Shift-Invariance)** For any graph $\mathcal{G}$ and shift component $\tau_i$, the encoder $h$ satisfies: $h(\tau_i(\mathcal{G})) = h(\mathcal{G})$
>
> **Proof:** Given the gating model's ability to identify shift components and the expert models' invariance properties:
>
> $$
> h(\tau_i(\mathcal{G})) = \sum_{j=0}^K w_j(\tau_i(\mathcal{G})) \cdot \xi_j(\tau_i(\mathcal{G}))
> = w_i(\tau_i(\mathcal{G})) \cdot \xi_i(\tau_i(\mathcal{G}))
> = w_i(\mathcal{G}) \cdot \xi_0(\mathcal{G})
> = h(\mathcal{G})
> $$
>
> The second equality holds because the gating model identifies $\tau_i$, and the third follows from the definition of referential invariant representation.
>
> **The distance term in Equation 3 regularizes**, enforcing invariance across shifts. This extends to combinations of shifts through our training objective and expert output alignment.
>
> **Our approach differs from simple ERM with data augmentation** by adaptively combining expert models to handle instance-specific shifts and ensuring invariance to individual and combined shift components.
>
> We sincerely appreciate the reviewer's astute observation and thank you for bringing this to our attention. We strive to make this clearer in the revision. We clearly distinguished between the standard definition of invariant representation and our concept of referential invariant representation, including lines 199 and 243, which are the only two instances in our methodology section. We also recognize that our statement (in an attempt to avoid confusion with new concepts) at lines 69-70 can be more precise, and we have revised it to accurately reflect the concept of referential invariant representation.
>
> We are grateful for the opportunity to address these points and present them better in our revision.
>
> **Related work**: This is interesting work! We see the intuitive similarity that both works move from specializing in a single environment to achieving invariance across multiple environments. The core difference is perhaps that they explicitly model domain correlations and spurious features, while we focus on experts that contribute to global invariance. We added it to related work as well. Thanks for bringing this relevant work to us!
>
> ---
>
> ## **Comment 2: Applying GraphMetro on molecular datasets**
>
> Thanks for the insightful comments. We provide answers from several angles:
>
> - **Adaptability to Molecular Graphs**:
>    GraphMETRO is designed to be adaptable with user-defined transform functions. For molecular graphs, we can incorporate domain-specific transform functions mimicking scaffold or assay shifts rather than relying solely on generic graph augmentations.
>
> - **Incorporating Domain Knowledge**:
>    As discussed in `Appendix G`, we can use domain knowledge to construct relevant transform functions. For molecular datasets, this could involve:
>    - Adding or modifying carbon structures while preserving functional groups.
>    - Altering specific molecular substructures relevant in scaffold shifts.
>    - Simulating changes in molecular properties that occur in assay shifts.
>
> - **Single Shift Scenario**:
>    GraphMETRO also works with a single shift where there will be an in-distribution expert (aka, reference model) and an OOD expert for one type of shift (e.g., scaffold shift). In a more flexible design, the gating model can be adapted/learned to focus on more relevant components since GraphMETRO doesn't require perfect alignment to be effective (c.f., `Theorem 3` for `reviewer ymW6`)
>
> - **Transfer Learning Approach**:
>    In `Appendix G`, we discussed a transfer learning setting leveraging a few samples from the target distribution to inform the selection or construction of transform functions with manual addition, ensuring better coverage of distribution shifts in molecular datasets.
>
> We believe these adaptations would be interesting future directions to apply GraphMETRO on molecular datasets. Thank you for highlighting it!
>
> ## **Summary**
>
> We are grateful for your positive evaluation on our novelty, applicability on both node and graph tasks, and effectiveness. We further provide clarifications on the invariant learning terminology and applicability on molecular data. We would very much appreciate it if you could reconsider your evaluation if some concerns are addressed. Thank you very much!

---

> > ### Comment · Reviewer_U8um · 2024-08-08
> >
> > Thank you for the authors' response. Your reply has given me a clearer understanding of how the proposed method works. The proposed _GraphMetro_ seems to be a variant of Distribution Robust Optimization (DRO), where different distribution shifts are not too far from the referential distribution to enforce model robustness. I have one more question: Does $k$ include $0$ in $\mathcal{L}\_2$?

---

> > > ### Author Response · Authors · 2024-08-08
> > > **About L_2 and DRO**
> > >
> > > Thanks for the quick reply! We're glad to know that our response has clarified the method!
> > >
> > > Yes, $k$ in $\mathcal{L}_2$ includes $k=0$. This mainly optimizes the reference model on the original training data with CE loss (since the second term in $\mathcal{L}_2$ is 0), making it an "in-distribution" expert.
> > >
> > > **GraphMETRO and DRO**: This is a thoughtful point! Yes both GraphMETRO and DRO enforce model robustness. One of the essential differences is perhaps on how they model shifts (a continuous mixture of shift components v.s. defined perturbation sets). Due to the infinite number of mixtures, standard DRO’s worst-case minimization may not be directly applicable, leading us to seek more tractable solutions in GraphMETRO.

---

> > > > ### Comment · Reviewer_U8um · 2024-08-10
> > > >
> > > > The author's response has effectively addressed my questions regarding how and why GraphMetro works, therefore I raise my score to 5. However, I still suggest that the authors make additional efforts to improve the clarity of the term "invariant" (e.g., in lines 75-76). This term may be misleading, as it refers to generating invariant representations with respect to referential representation, rather than representations that remain stable across different environments.

---

> > > > > ### Author Response · Authors · 2024-08-10
> > > > > **Thank you**
> > > > >
> > > > > Thank you for your feedback and for raising the score!
> > > > >
> > > > > To sum up, GraphMETRO's objective is for the experts to output referentially invariant representations, and further for the final MoE model to output invariant representations.
> > > > >
> > > > > We have made explicit clarifications in the current revision, which should avoid any confusion regarding these distinctions. Thank you again for your constructive comments!

---

### Author Rebuttal · Authors · 2024-08-06

# **General response**

We truly appreciate the reviewers' efforts and valuable suggestions in reviewing our paper. We are glad that most reviewers reached a positive consensus on our work's motivation, presentation, novelty, and experimental effectiveness. Since we received a decent number of reviews, we provide a summary on the reviewers’ major feedback and our corresponding actions:

| | Reviewer U8um| Reviewer uY6m| Reviewer LHfk| Reviewer t31K| Reviewer ymW6| [Action] Referring to existing content | [Action] Modification |
|:---|:---|:---|:---|:---|:---|:---|:---|
|**Motivation**| NA | “problem is interesting…motivation is meaningful”| NA | NA | “This paper studies an interesting research problem” | `NA`|`NA`|
|**Presentation**|“Presentation: good” | “Presentation: good” | “well-written and provides a detailed experimental analysis.” | “Presentation: good”| “The model design is easy to understand” |`NA`|`NA`|
|**Novelty**|”This work proposes a novel idea”|“The model design is novel and reasonable… good interpretability.”|”proposes a novel method to addresses distribution shifts” | “a framework that is able to decompose any distribution shift into multiple shift components.” | “....instance heterogeneity for graph OOD problems is not new…the authors could address my concern by presenting more differences with related works” | `NA` | RE Reviewer ymW6: `We justify our novelty with extensive comparison with related works` |
|**Empirical Improvements**|”can achieve significant performance improvement…such as WebKB.”| “Extensive experiments verify the effectiveness” | “This empirical validation demonstrates the effectiveness and applicability” |”achieves state-of-the-art results on real-world GOOD benchmarks and provides interpretability…”| “The performance improvement on some comparisons seems to be significant.” | `NA`|`NA`|
|**Applicability**| ”...can be applied for both node and graph-level classification”; Question about applying GraphMETRO on molecular graphs| “The actual distribution shift can be very complex” | “The paper proposes five types of graph distribution shifts…but real-world graph distribution shifts may be more varied” | NA | NA | `We point the reviewers to Appendix G, where we had discussed the applicability of GraphMETRO including on molecular datasets.` | `NA` |
| **Clarification**| (1) Definition of “invariance” and (2) Objective (Equation 3) | Suggest to improve notations | NA | NA | NA | `NA`  |RE Reviewer U8um: `We provide in-depth discussions on "invariance" and our objective`; RE Reviewer uY6m: `We improved our notations`;  |
| **Extended Evaluation**  | NA |NA | Suggest to add the latest OOD methods and more datasets  | NA| NA| `NA` | `We include two more baselines: (1) OODGAT suggested by the reviewer, and (2) G-Mixup (ICML 2022)`|
| **Property of GraphMETRO**  | NA | NA | (1) Impact (ablation) of aligning expert representations | (1) Impact of aligning expert representations, (2) Whether the expert model can consistently produce invariant features, (3) The effects of different k values on the framework’s performance|Request on detailed theoretical analyses| RE Reviewer LHfk and t31K: `We point the reviewers to` (2) `Section 4.3, where we validated the invariance of the embeddings produced by the expert models`, and (3) `Appendix E where we study on the effect of the $k$ value` | RE Reviewer ymW6: `We provide more detailed theoretical analysis` RE Reviewer LHfk and t31K: (1) `We conduct ablation study to measure the impact of alignment`;  |

------


# **Summary**

- We address comments on the applicability and properties of GraphMETRO by referring to our existing content.
- For the other concerns, we mainly made modifications to the experiments and comparisons with recent graph OOD methods.

We thank the reviewers for their suggestions, which have made our work more solid. We hope our responses clarify any confusion and alleviate the concerns.

We would be thrilled if you could let us know whether your concerns have been addressed or if you have any follow-up questions!

— Best,

  Authors of Paper 5517

---

### Decision · Program_Chairs · 2024-09-25

**Decision:**

Accept (poster)

**Comment:**

This paper proposes GraphMETRO, a graph neural network architecture designed to handle complex distribution shifts in graph data. It uses a mixture-of-experts approach with multiple expert models targeting specific distribution shifts and a gating model to identify shifts. The goal is to produce shift-invariant representations that generalize well to out-of-distribution data.

The reviewers unanimously agree to accept this paper provided that the additional experiments and theoretical analyses are included in a camera-ready version of the paper. Some reviewers further have suggestions to improve notations and the overall writing.
Overall, the authors have addressed the major concerns raised by reviewers and the paper appears to make a solid contribution to graph OOD generalization. The mixture-of-experts approach is novel and the empirical results are strong.